# Live and heat-killed *Leuconostoc mesenteroides* counteract the gastrointestinal dysfunction in chronic kidney disease mice through intestinal environment modulation

**Fittree Hayeeawaema[1], Natthawan Sermwittayawong[2], Chittipong Tipbunjong[1], Nawiya Huipao[1], Paradorn Muangnil[3], Pissared Khuituan[1]\***

**1** Division of Health and Applied Sciences, Faculty of Science, Prince of Songkla University, Hat Yai, Thailand, **2** Division of Biological Science, Faculty of Science, Prince of Songkla University, Hat Yai, Thailand, **3** Faculty of Veterinary Science, Prince of Songkla University, Hat Yai, Thailand

\* pissared.k@psu.ac.th

## Abstract

Probiotics are well-known therapeutic agents for managing constipation and have been used to improve chronic kidney disease (CKD) progression. However, heat-killed probiotics on CKD remain inadequately explored. This study aimed to evaluate the probiotic potential of lactic acid bacteria derived from natural sources and to investigate the effects of both live and heat-killed *Leuconostoc mesenteroides* (Ln.m) on renal and gastrointestinal functions in CKD mice. Ln.m was selected from acid and bile salt intolerance tests, non-hemolytic activity, and antibiotic sensitivity. CKD mice demonstrated significantly elevated blood urea nitrogen (BUN) and creatinine levels compared to control mice ($p < 0.001$ and $p < 0.01$). Treatment with live and heat-killed Ln.m significantly reduced BUN and creatinine levels in CKD mice ($p < 0.01$ and $p < 0.05$). Additionally, kidney damage observed in CKD mice compared to control mice, including glomerular necrosis, tubular dilatation, inflammation, and fibrosis, was significantly alleviated following live and heat-killed Ln.m treatments. CKD-induced gastrointestinal dysfunction was characterized by an imbalance in Firmicutes/Bacteroidota populations, increased colonic uremic toxin ($p < 0.01$), reduced fecal short-chain fatty acids (SCFAs) ($p < 0.05$), and constipation. Treatment with live and heat-killed Ln.m restored gut microbiota, decreased uremic toxin ($p < 0.001$), increased SCFAs ($p < 0.05$), and alleviated constipation. In summary, both live and heat-killed Ln.m effectively alleviated gastrointestinal dysfunction and renal damage in CKD mice, primarily through modulation of the intestinal environment. These findings highlight the therapeutic potential of live and heat-killed Ln.m as the gastrointestinal dysfunction treatment in CKD.

## Introduction

Chronic kidney disease (CKD) is a progressive condition characterized by the gradual loss of kidney function. The kidneys play essential roles in maintaining fluid, acid, and mineral balance and eliminating metabolic waste products and toxins. Consequently, the decline in kidney function leads to metabolic waste and toxins accumulation in the body [1]. CKD is a

provided the original author and source are credited.

**Data availability statement:** All relevant data are within the manuscript and its Supporting information files.

**Funding:** This research is supported by a grant from the National Research Council of Thailand (NRCT): NRCT5-RGJ63019-160. The funders had no role in study design, data collection and analysis, decision to publish, or preparation of the manuscript.

**Competing interests:** The authors have declared that no competing interests exist.

significant global health concern, as its advanced stages can result in severe complications, including cardiovascular diseases, anemia, and gastrointestinal disorders [1,2]. In CKD patients, waste products such as urea accumulate in the systemic circulation and enter the gastrointestinal lumen, disrupting the gut microbiota in a condition known as uremic dysbiosis [3]. This dysbiosis represents an imbalance between the two dominant phyla of gut microbiota: Firmicutes and Bacteroidetes (Bacteroidota). CKD patients exhibit reduced gut microbiota diversity and bacterial abundance, with a marked increase in the proportion of Firmicutes [4]. Similarly, in 5/6 nephrectomy (Nx) rats, the abundance of *Bacteroides* increased, while *Lactobacillus* populations decreased. Both bacterial species were significantly associated with urinary protein excretion [2]. Moreover, short-chain fatty acids (SCFAs), the key microbial metabolites such as acetic, propionic, and butyric acids were significantly reduced alongside the declining diversity of gut microbiota in CKD mice [5]. Reduced SCFAs production has been identified as a critical factor contributing to intestinal dysmotility, including constipation, in CKD patients [1].

Constipation is the most common gastrointestinal complication associated with CKD and is influenced by multiple factors, including dietary restrictions, medications, and comorbidities such as diabetes. These factors collectively contribute to alterations in gut microbiota [6]. While constipation itself is not life-threatening, it significantly accelerates the progression of CKD to end-stage renal disease (ESRD) [7]. CKD patients with constipation are at a substantially higher risk of developing ESRD compared to those without constipation [6]. Several mechanisms link constipation to CKD progression. Prolonged intestinal transit exacerbates gut microbiota dysbiosis, leading to an increased production of gut-derived uremic toxins such as indole and p-Cresol. These toxins are absorbed into the bloodstream and metabolized into indoxyl sulfate and p-Cresol sulfate, respectively. Normally, they are excreted by the kidneys, but these compounds accumulate in the circulation in CKD patients, contributing to kidney fibrosis and further disease progression [6–8]. Given its significant impact, constipation is a critical complication in CKD that necessitates effective therapeutic strategies with minimal side effects. Alongside pharmacological treatments, non-pharmacological approaches including exercise, prebiotics, and probiotics have demonstrated beneficial effects in managing constipation and supporting kidney health in CKD patients [9].

The Food and Agriculture Organization (FAO) and World Health Organization (WHO) define probiotics as "live microorganisms that, when administered in adequate amounts, confer a health benefit on the host." Most probiotics belong to the lactic acid bacteria (LAB) group and are primarily derived from the healthy human gastrointestinal tract [10]. Probiotics have shown promising effects in managing CKD. In adenine-induced CKD rats, four weeks of probiotic treatment improved gut microbiota dysbiosis, increased SCFAs production, and reduced intestinal permeability and uremic toxin levels [11]. Similarly, treating nephrectomy (Nx) rats with a *Lactobacillus* mixture not only improved blood urea nitrogen (BUN) and creatinine levels but also mitigated gut dysbiosis, glomerular sclerosis, and kidney fibrosis and effectively reversing kidney damage [2]. Furthermore, dietary supplementation with probiotics has been reported to decrease BUN, indoxyl sulfate, and p-cresyl sulfate levels, underscoring their potential as an intervention in both CKD animal models and clinical trials [3,12,13]. Probiotics exert their beneficial effects on CKD through several mechanisms: i) modulating gut microbiota composition, ii) antagonizing pathogenic bacteria by releasing bacteriocins and competing for nutrients and adhesion sites, iii) enhancing epithelial integrity by increasing mucus secretion and strengthening epithelial tight junctions, and iv) increasing the production of SCFAs, which play a crucial role in gut and kidney health. These findings highlight the therapeutic potential of probiotics in addressing CKD-related complications through multiple pathways [12].

Some probiotics, such as *Bifidobacterium longum* and *Lactobacillus acidophilus*, have been isolated from human feces, while others originate from animal gastrointestinal tracts [14]. With the increasing use of probiotics and the growing probiotic market, numerous lactic acid bacteria (LAB) strains with probiotic properties have been developed from natural sources. To be considered probiotics, these bacteria must demonstrate safety, tolerance to acid and bile salts, and provide measurable health benefits, such as anti-cancer effects, gut microbiota modulation, anti-microbial activity, or inhibition of toxin production [15]. In addition to live probiotics, inactivated probiotics, also known as paraprobiotics, offer the potential for CKD improvement [3]. Paraprobiotics are defined as "inactivated non-viable microbial cells or cell fractions that confer a health benefit to the consumer" [16]. These inactivated forms can avoid the potential risks of live probiotics, such as translocation from the gastrointestinal tract to the bloodstream, which is particularly relevant for vulnerable populations, including pediatric or immunocompromised patients. Furthermore, paraprobiotics are easier to transport, store, and standardize compared to their live counterparts [17]. Previous studies have demonstrated that inactivated probiotics can retain their health benefits. For example, heat-killed (HK) *Lactobacillus reuteri* GMNL-263 was shown to reduce insulin resistance and hepatic steatosis in high-fat diet-induced obese rats. Similarly, HK *Lactiplantibacillus plantarum* (HKLp-nF1) improved defecation status, intestinal health, and cytokine levels in loperamide-induced constipated rats, with effects comparable to live probiotics [18,19]. Both live and HK probiotics have shown positive effects on gastrointestinal health across multiple studies; however, the specific effects of HK probiotics in CKD remain underexplored.

Thus, this study aims to investigate the impact of both live and HK *Leuconostoc mesenteroides* (Ln.m) on gastrointestinal functions in CKD mice, providing new insights into the potential applications of heat-killed probiotics in managing CKD.

## Materials and methods

### Screening of isolated LAB from the gastrointestinal tract of shrimp

**Bacterial strains and growth condition.** To isolate the LAB from different natural sources, fermented cabbage, fermented garlic, Sea bass's gastrointestinal content, and Tiger prawn's gastrointestinal content were collected under hygienic conditions. The samples were homogenized and incubated in the De Man, Rogosa, and Sharpe (MRS) broth at 37°C with shaking at 150 revolutions per minute (RPM) for 24 h. Afterward, the serial dilution was performed before plating on the MRS agar and incubated at 37°C for 24 h in the anaerobic condition. The colonies were selected from their different morphology. The preliminary experiment was performed by acid tolerance test and found that only the LAB from Tiger prawn gastrointestinal content could survive. Ten LAB strains, P.6.1, P.6.4, P.6.6, P.7.1, P.7.2, P.7.4, P.7.5, P.9.1, P.10.1, and P.11.4, were selected and re-cultured in MRS broth at 30°C with shaking at 150 RPM for 24 h. In this study, *Lactiplantibacillus plantarum* (*L. plantarum*) ATCC 14917 (TISTR 877) and *Lactiplantibacillus casei* (*L. casie*) ATCC 7469 (TISTR 047) were used as probiotic standards.

**Acid and bile salt tolerance.** To test the acid tolerance, 1% of $10^9$ CFU/mL culture was transferred into MRS broth at pH 7 (control), pH 4, or pH 3, and incubated at 37°C with shaking at 150 rpm for 0, 1, 2, and 3 h. To test bile salt tolerance, the cultured was transferred into MRS broth containing 0, 0.2, and 0.3% (w/v) of bile salt (Ox gall; conjugated bile salt), and incubated at 37°C with shaking at 150 rpm for 0, 1, 2, and 3 h. Serial dilution was performed at each time point, and the surviving bacteria were enumerated on the MRS agar plate as biomass (CFU/mL). The assays were done in 3 replicates. The survival rate was calculated as survival rate (%) = biomass at time/biomass at time 0 x 100.

**Antibiotic susceptibility.** The susceptibility test was performed by the disc diffusion method [20]. Overnight cultures of the isolated strains in MRS broth were adjusted to $10^9$ CFU/mL and spread on Mueller Hinton agar. The following antibiotic discs (Oxoid) were used: amikacin (30 μg), ampicillin (10 μg), ceftriaxone (30 μg), chloramphenicol (30 μg), ciprofloxacin (5 μg), clindamycin (2 μg), erythromycin (15 μg), gentamycin (10 μg), kanamycin (30 μg), streptomycin (10 μg), sulbactam (20 μg), tetracycline (30 μg), and vancomycin (30 μg). Antibiotic discs were placed on the agar after the spread strain was dry. The plates were incubated at 37°C for 24 h in the anaerobic condition. Inhibition zones around antibiotic discs were measured in millimeters (mm). The sensitivity or resistance was indicated following the previous studies [21–23].

**Hemolytic activity.** Hemolysis was tested according to Somashekaraiah et al. (2019) [15]. The LAB strains were streaked onto blood agar with 5% (w/v) blood. Hemolytic patterns were observed after 37 °C incubation for 48 h in the anaerobic condition. The results were described as β-hemolysis (complete hemolysis) which appeared as a transparent zone surrounding the colony, α-hemolysis (partial or incomplete hemolysis) which appeared as a greenish zone surrounding the colony, and γ-hemolysis when the medium around the colony remained unchanged.

**Inactivation of LAB.** Heat inactivation of bacteria was performed based on the previous methods with slight modifications [18,24]. The selected LAB strain was grown in MSR broth in the anaerobic condition at 37 °C with shaking at 150 rpm for 24 h. The bacteria were harvested by centrifugation at 8,000 rpm for 10 min and washed twice with 0.85% NaCl [15]. Bacterial cells were resuspended in 0.85% NaCl and then inactivated in the water bath at 100 °C for 10 min. The HK bacteria were freeze-dried and kept at –80 °C until used. The freeze-dried bacteria powder was suspended in sterile distilled water to be administered to the mice.

## Effects of selected LAB on CKD mice

**Animals.** Male ICR mice (7–8 weeks old) were purchased from Nomura Siam International Co., Ltd., Thailand, and housed at the Southern Laboratory Animal Facility, Prince of Songkla University, Thailand. Animals were housed under the controlled environmental standard: a 12 h light/ dark cycle, at 23–27 °C, in 50–55% humidity, with free access to the standard chow and water *ad libitum*. Mice were allowed to adapt to the environment for a week before the experiments. The mice were divided into six experimental groups (10 animals/group) as follows: 1. Control: healthy mice receiving distilled water, 2. CKD + water: CKD-induced mice receiving distilled water, 3. CKD + LLp: CKD-induced mice receiving $2 \times 10^9$ CFU live *L. plantarum* (LLp), a standard probiotic (ATCC 14917), 4. CKD + HKLp.: CKD-induced mice receiving $2 \times 10^9$ CFU HK *L. plantarum* (HKLp), 5. CKD + LLn.m: CKD-induced mice receiving $2 \times 10^9$ CFU live *Leuconostoc mesenteroides* (LLn.m) (the selected LAB), and 6. CKD + HKLn.m: CKD-induced mice receiving $2 \times 10^9$ CFU HK *Leuconostoc mesenteroides* (HKLn.m). To induce CKD, mice in all groups (except the control group) were intraperitoneally (i.p.) injected with 50 mg/kg adenine once a day for 28 days. After CKD induction, mice were orally administered LLp, HKLp, LLn.m, or HKLn.m once a day (daily) for 28 days. Freeze-dried live and HK probiotics were freshly re-suspended in sterilized distilled water before oral gavage, with each mouse receiving 0.2 mL/day. Body weight (BW), food and water intake, and general health were monitored every day during the experimental period. At the end of the experiment, mice were anesthetized by i.p. injection with 70 mg/kg thiopental sodium for sample collection and were sacrificed by cervical dislocation [25]. All experimental procedures in this study were approved by the Animal Ethics Committee of Prince of Songkla University, Thailand (Ethical clearance MHESI 6800.11/911).

**Biochemical and complete blood count parameters.** Blood samples were collected by cardiac puncture from mice under deep anesthesia. The complete blood count (CBC) analysis was performed using BC-2800Vet Auto Hematology Analyzer (Mindray, Shenzhen, China). CBC analysis parameters measured included white blood cells (WBC), lymphocytes, monocytes, granulocytes, red blood cells (RBC), hemoglobin (HGB), hematocrit (HCT), mean corpuscular volume (MCV), mean corpuscular hemoglobin (MCH), mean corpuscular hemoglobin concentration (MCHC), platelets (PLT), and mean platelet volume (MPV). Whole blood samples were centrifuged at 4,000 rpm for 10 min to separate plasma. Plasma levels of BUN, creatinine, aspartate transaminase (AST), alanine transaminase (ALT), and alkaline phosphatase (ALP) were analyzed using the BS-20 Chemistry Analyzer (Mindray, Shenzhen, China).

**Gut microbiota population.** The colonic content of mice was collected in sterile conditions and genomic DNA was extracted. The 16S rRNA genes of the gut microbiota population were amplified by PCR at the V3-V4 region [26]. The amplicons were purified with a PCR product purification kit (Illumina). The samples were library-prepared and qualified for sequencing by Novaseq sequencing. The cluster sequences were denoised by the DADA2 method. The generated Amplicon Sequence Variants (ASVs) were analyzed to identify common and unique ASVs between samples. The results were presented as a flower diagram. Beta diversity represents the obvious microbial community based on the diversity of the microbial composition indicated by the species and abundance in the ASV data. Beta diversity was analyzed using Unifac distance and the differences among samples were analyzed by Principal Coordinate Analysis (PCoA) [27].

**Uremic toxins.** Collected colonic contents were kept at –80 °C until used. Samples were dissolved in phosphate-buffered saline at 1:9 (w/v) and mixed before centrifuging at 10,000 rpm for 10 min. The supernatant was collected and filtered through a 0.2 μm nylon filter. The concentration of indole and p-Cresol were measured by high-performance liquid chromatography (HPLC), using a $4.0 \times 250$ mm Agilent Hypersil ODS C18 column (USA). The UV detector and column temperature were set at 220 nm and 35 °C. Mobile phases A and B were 50% Milli-Q water and 50% acetonitrile, flowing at 0.5 mL/min. SCFAs produced by microbial fermentation, including acetate, propionate, lactate, and butyrate, were analyzed by HPLC, using a $300 \times 8.0$ mm Shodex SUGAR SH1011 column (Tokyo, Japan). The UV detector and column temperature were set at 215 nm and 50 °C. The eluent solvent was 5 mM $H_2SO_4$, flowing at 0.5 mL/min [28]. Concentrations of uremic toxins and SCFAs were calculated by area of peak compared to the standard control using Chemstation software (Version CHEM32, USA).

**Constipation.** Constipation was determined by decreasing defecation frequency, fecal water content, and gastrointestinal transit. To record the frequency of defecation, feces were collected for 4 h and fecal pellets were counted as the number of defecations. To measure fecal water content, collected wet feces were weighed and then dried at 100°C for 30 min before being weighed dry. The fecal water content (%) was calculated by:

$$\text{Fecal water content}(\%) = \frac{\left(\text{wet weight} - \text{dry weight}\right)}{\text{wet weight}} \times 100$$

The upper gastrointestinal transit was calculated by using the charcoal meal (10% activated charcoal and 5% gum acacia) method. Mice were orally administered 0.2 mL charcoal meal for 30 min before anesthetized. The distance of the charcoal marker traveled and the total small intestinal length was measured with cotton thread immediately after anesthetization and abdominal incisions were made. Gastrointestinal transit (%) was calculated by:

$$\text{Gastrointestinal transit}(\%) = \frac{\text{distance of charcoal meal marker}}{\text{total length of the small intestine}} \times 100$$

**Intestinal smooth muscle contractility.** After anesthetized, the ileum and distal colon were sectioned and immediately placed in cold Krebs solution. A 1 cm of tissue was suspended in an organ bath containing Krebs solution. The temperature of the bath was maintained at 37 °C with continuously oxygenated with carbogen. Each tissue was placed under initial tension at 0.5 g before the contractility was recorded. Smooth muscle contractions were detected by a force transducer (Model FT03, Grass, MA, USA), recorded by the PowerLab System (AD Instruments, New South Wales, Australia), and analyzed by LabChart7 software. Smooth muscle contractility was determined by the motility index as described by Hoibian et al., 2018 [29] and calculated as Ln((number of peaks × sum of peak amplitudes) + 1).

**Kidney and colon histopathology.** After euthanized, the kidney and intestine were removed and fixed in 10% formalin for 24 h. Tissues were dehydrated in graded concentrations of ethanol, embedded in paraffin, cross-sectioned at 5 μm in the paraffin block, and stained with hematoxylin and eosin (H&E) or Masson's Trichrome, following standard protocols. Stained sections were examined under a light microscope (Olympus DP73). Glomerular necrosis was determined by Bowman's capsule space, distal tubular dilatation was determined by the diameter of the tubule, and the frequency of inflammation was determined by the appearance of cellular infiltration. Kidney fibrosis was expressed as the area of fibrosis (%) which was determined by measuring the red threshold from 0–120 in the Image J software [30]. Intestinal pathologies, such as abnormal morphology, inflammation, and fibrosis in the submucosal layer were also determined.

**Statistical analysis.** The data were presented as means ± the standard error of the mean (SEM). One-way analysis of variance (ANOVA) followed by the Bonferroni post hoc test was used for significant differences analysis. Data were analyzed using GraphPad Prism version 5 (GraphPad Software Inc., San Diego, CA, USA). A p-value < 0.05 was set as statistically significant.

## Results

### Acid and bile salt tolerances of LAB

The acid and bile salt tolerance tests evaluated the ability of isolated LAB strains to withstand the acidic conditions of the stomach and bile salt concentrations of the small intestine, ensuring their potential to reach the colon. Results indicated that LAB strains in control conditions (pH 7, 0% bile salt) showed a gradual increase in survival rates over 3 h, reflecting their normal growth (Figs 1A and 1D). At pH 4 and pH 3 with 0% bile salt, most strains exhibited survival rates exceeding 90%, except for strains P.7.2, P.10.1, and P.11.4, which showed survival rates dropping below 60% at pH 3 (Figs 1B and 1C). In the presence of bile salts, only *L. plantarum*, P.6.1, and P.6.6 demonstrated significant tolerance. Both *L. plantarum* and P.6.6 maintained survival rates exceeding 60% in 0.2% bile salt, while P.6.1 was entirely inhibited under the same conditions after 3 hours (Fig 1E). At 0.3% bile salt, only *L. plantarum* and P.6.1 achieved survival rates above 60%, but neither could persist beyond 2 hours (Fig 1F).

### Antibiotic susceptibility of LAB

All isolated strains were sensitive to ampicillin, chloramphenicol, erythromycin, sulbactam, and tetracycline but resistant to amikacin, aztreonam, ciprofloxacin, gentamycin 10, kanamycin, streptomycin, sulfamethoxazole, vancomycin. All sensitive except *L. plantarum* were resistant to cefoxitin and clindamycin (S1 Table).

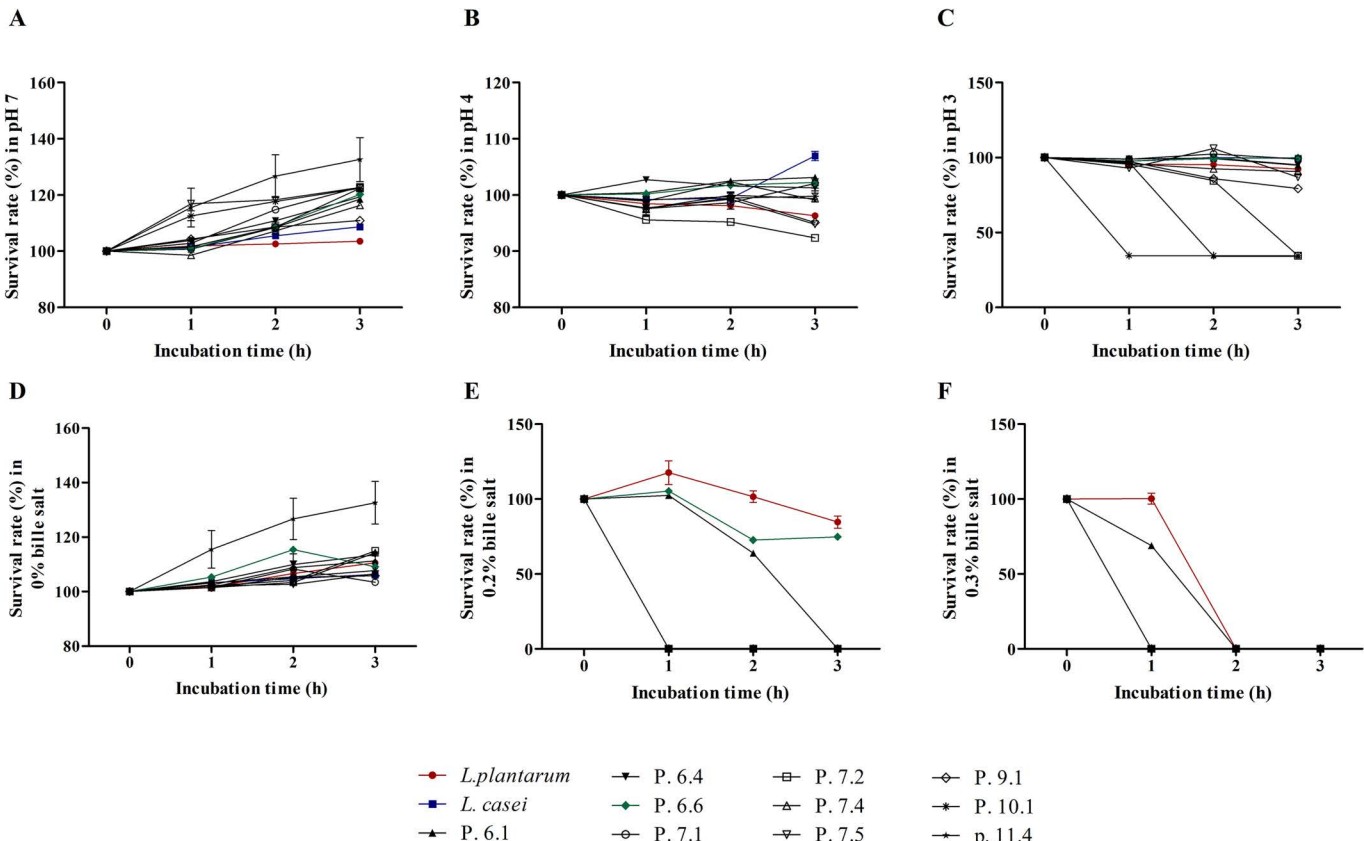

**Fig 1. Acid and bile salt tolerances of isolated lactic acid bacteria (LAB).** A, B, and C: The survival rate (%) in acid when LAB was incubated at pH 7 (control), pH 4, and pH 3 for 0, 1, 2, and 3 h, respectively. D, E, and F: The survival rate (%) in bile salt when LAB was incubated in 0% (control), 0.2%, and 0.3% bile salt for 0, 1, 2, and 3 h, respectively. Data were shown as means ± SEM (n = 3). *L. plantarum*; *Lactiplantibacillus plantarum*, *L. casei*; *Lactobacillus casei*.

## Hemolytic activity of LAB

All the isolated LAB strains showed no hemolytic activity, except P.7.5, which produced a transparent zone under the colony. Therefore, P.7.5 exhibited complete hemolysis (β–hemolysis). Taken together, most isolated LAB was tolerant to low pH for 3 h of incubation, while only P.6.6 was tolerant for 3 h of incubation in 0.2% bile salt. Even though P.6.1 could tolerate 0.3% bile salt, it could do so only for 2 h of incubation. The antibiotic susceptibility test showed most of the LAB were sensitive to antibiotics. Some resistance was observed; *L. plantarum* resisted ceftriaxone, *L. plantarum*, *L. casei*, P.7.1, and P.7.4 resisted kanamycin, P.7.4 resisted gentamicin, *L. casei* and P.7.1 resisted streptomycin, and all antibiotics strains, except P.7.1, and P.7.4, resisted vancomycin.

P.6.1 and P.6.6 showed similar antibiotic-resistant profiles, but P.6.6 was more susceptible to antibiotics than P.6.1. Therefore, P.6.6 was selected as the probiotic for further study. Before using P.6.6 as a probiotic, 16S rDNA gene sequencing was performed to identify the bacterial species. Sequence analysis identified P.6.6 as *Leuconostoc mesenteroides*.

## Effects of live and HK Ln.m on body weight, and water and food intakes in CKD mice

The gradual increase in body weight (BW) observed in the control group throughout the experimental period reflected normal growth. In contrast, mice in the CKD group showed

a significant reduction in BW compared to the control group from day 21 to day 28 of the induction period and up to day 7 of the treatment period ($p < 0.05$). However, no significant differences in BW were detected between the CKD probiotic treatment groups and the untreated CKD group (Fig 2A).

Water intake in the CKD group was significantly higher than in the control group from day 7 to day 28 of the induction period ($p < 0.001$) and from day 7 to day 14 of the treatment period ($p < 0.001$ and $p < 0.05$). No significant differences in water intake were observed between the CKD probiotic-treated groups and the CKD group (Fig 2B). Although food intake was lower in the CKD and CKD probiotic-treated groups compared to the control during the induction period, the differences were not statistically significant (Fig 2C). In summary, water intake in the CKD group increased significantly from the onset of CKD induction through the treatment period, while BW decreased, with no change in food intake. These findings suggest that CKD mice experienced higher glomerular filtration rates and increased water excretion during and after CKD induction.

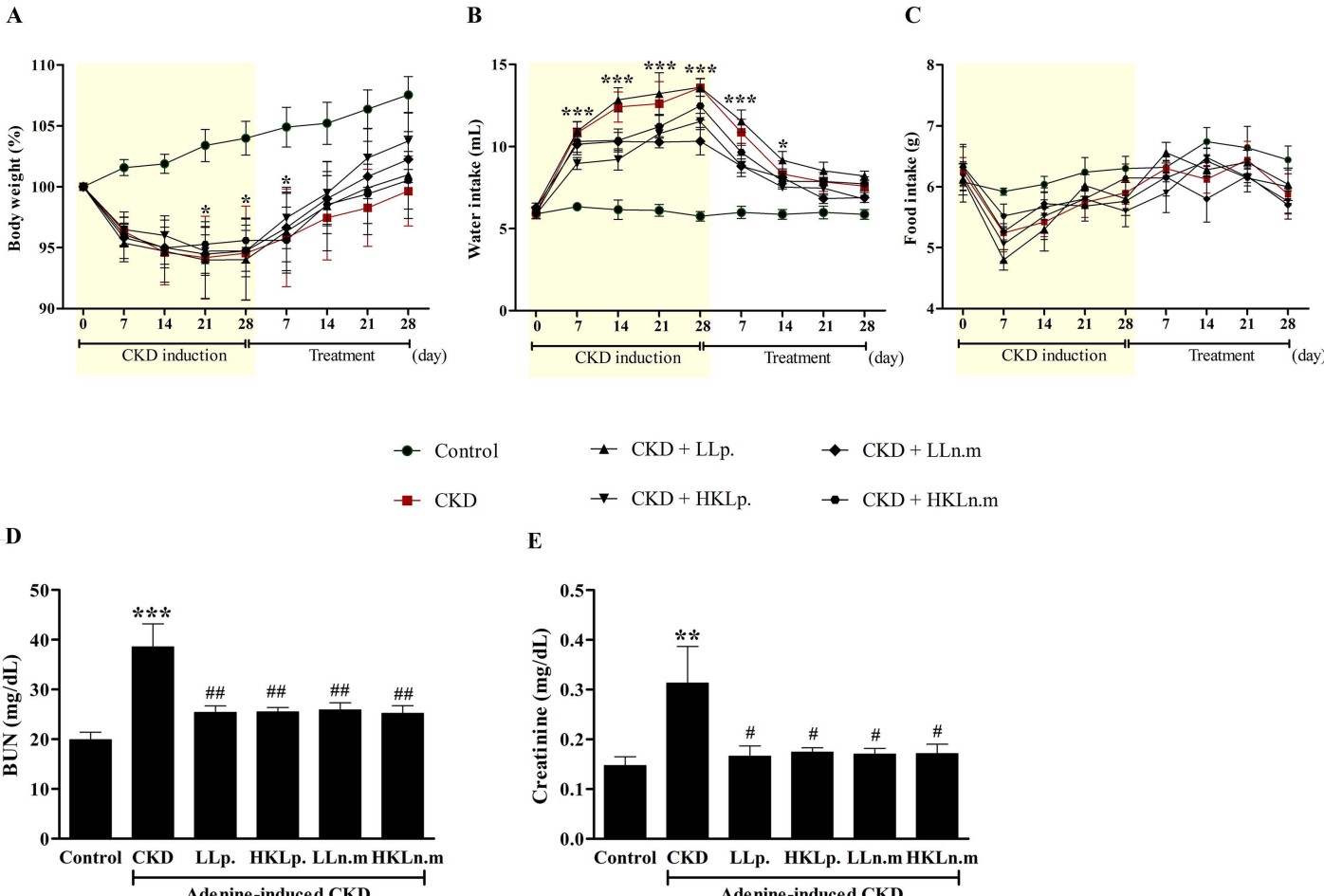

**Fig 2. Effects of live and heat-killed *Leuconostoc mesenteroides* on body weight, water intake, food intake, blood urea nitrogen (BUN), and creatinine levels in chronic kidney disease (CKD) mice.** A, B, and C: Body weight (%), water intake (mL), and food intake (g) of mice during 28 days of CKD induction and 28 days of probiotic treatments, respectively. D and E: BUN (mg/dL) and plasma creatinine levels (mg/dL) of mice after treatments. Data were shown as means ± SEM (n = 5–10). *$p < 0.05$, **$p < 0.01$ and ***$p < 0.001$ when compared to the control group; #$p < 0.05$ and ##$p < 0.01$ when compared to the CKD group.

### Effects of live and HK Ln.m on BUN and plasma creatinine levels in CKD mice

At the end of the experiment, BUN and plasma creatinine levels were measured to evaluate the severity of CKD. BUN levels in the CKD group (38.60 ± 4.67 mg/dL) were significantly higher than in the control group (19.92 ± 1.50 mg/dL) ($p < 0.001$). In contrast, CKD mice treated with LLp, HKLp, LLn.m, and HKLn.m exhibited significantly lower BUN levels compared to the untreated CKD group ($p < 0.01$) (Fig 2D). Similarly, plasma creatinine levels were significantly elevated in the CKD group (0.31 ± 0.07 mg/dL) compared to the control group (0.14 ± 0.02 mg/dL) ($p < 0.01$). Treatment with LLp, HKLp, LLn.m, and HKLn.m significantly reduced plasma creatinine levels relative to the untreated CKD group ($p < 0.05$) (Fig 2E). These findings suggest that both live and HK Ln.m have beneficial effects on reducing CKD severity, comparable to those of a standard probiotic.

### Effects of live and HK Ln.m on hematological parameters and liver function in CKD mice

No significant differences were observed in any hematological parameters, indicating the absence of anemia or systemic inflammation in the CKD mice (S2 Table). Liver function was assessed by measuring liver enzyme levels. While no significant differences were observed in AST and ALT levels between the groups (S1A and S1B Figs), ALP levels were significantly higher in the CKD group (67.36 ± 5.17 U/L) compared to the control group (47.61 ± 5.89 U/L) ($p < 0.05$), suggesting liver damage or infection in the CKD group. Treatment with probiotics resulted in lower ALP levels in the CKD groups compared to the untreated CKD group. Notably, the reduction in ALP levels was significant in CKD mice treated with live and HK Ln.m ($p < 0.01$ and $p < 0.05$), indicating a potential reduction in liver damage in these groups (S1C Fig).

### Effects of live and HK Ln.m on gut microbiota population in CKD mice

A total of 107 common ASVs were shared among the six samples. The CKD group exhibited only 72 unique ASVs, compared to 224 unique ASVs in the control sample. Probiotic-treated groups demonstrated higher numbers of unique ASVs compared to the CKD group, with 138, 309, 253, and 136 unique ASVs observed in CKD.LLp, CKD.HKLp, CKD.LLn.m, and CKD.HKLn.m, respectively (Fig 3A). These findings suggest reduced gut microbiota diversity in the CKD group relative to both the control and probiotics-treated groups.

The PCoA plot revealed that PC1 and PC2 accounted for 73.95% and 11.48% of the variation in sample composition. The CKD sample was distinctly separated from the control and treatment groups. Notably, CKD.LLn.m and CKD.HKLn.m exhibited microbial compositions similar to the control group, while CKD.LLp and CKD.HKLp displayed less similarity to the control but remained distinct from the CKD group (Fig 3B).

The phylum- and genus-level composition and proportions in each sample were analyzed based on ASV data, revealing differences in taxonomic levels among groups. The percentages of bacteria from the phylum *Firmicutes* in the control, CKD, CKD + LLp, HKLp, LLn.m, and HKLn.m groups were 45.77%, 51.49%, 44.86%, 47.53%, 45.03%, and 46.23%, respectively. Bacteria from the phylum *Bacteroidota* (*Bacteroides*) accounted for 40.72%, 20.83%, 37.65%, 37.40%, 35.95%, and 36.42% of the bacterial population in these groups, respectively. The phylum *Verrucomicrobiota* was notably more abundant in the CKD group (17.54%) compared to other groups (Fig 3C). At the genus level, the relative abundance of *Muribaculaceae* was lower in the CKD group than in other groups, whereas the relative abundances of *Lactobacillus* and *Akkermansia* were higher in the CKD group. Although CKD.LLp showed a trend toward reversing the effects of CKD, it displayed a bacterial genera profile similar to the CKD group, indicating that it did not effectively restore the microbiota composition to normal levels (Fig 3D).

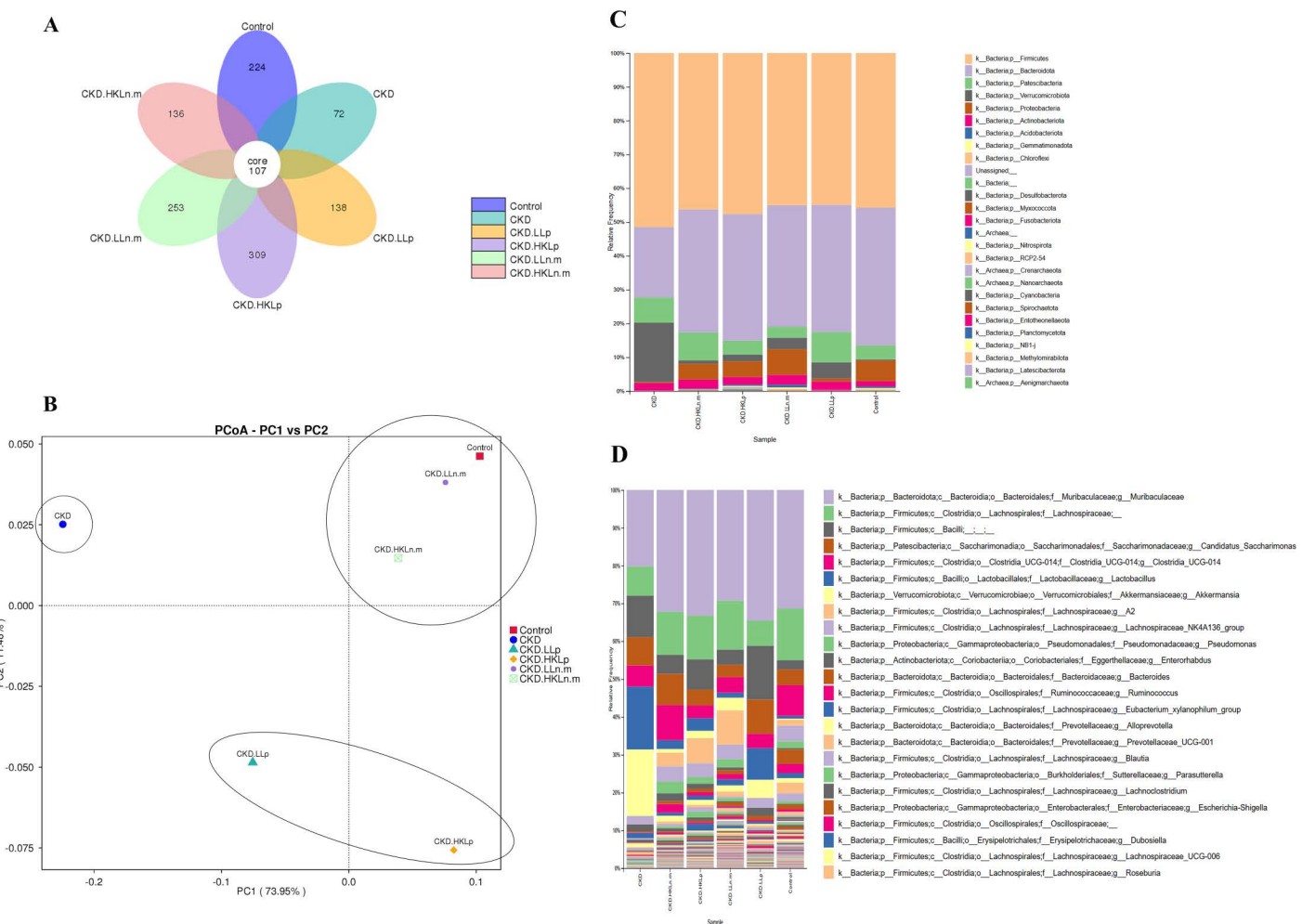

**Fig 3. Effects of live and heat-killed *Leuconostoc mesenteroides* on gut microbiota populations in chronic kidney disease (CKD) mice.** A: The flower diagram shows the common ASVs shared in six samples. B: PCoA plot showing PC1 and PC2 determination in six samples. C and D: Bacteria at phylum and genus levels. CKD; chronic kidney disease, LLp; live *Lactiplantibacillus plantarum*, HKLp; heat-killed *Lactiplantibacillus plantarum*, LLn.m; live *Leuconostoc mesenteroides*, and HKLn.m; heat-killed *Leuconostoc mesenteroides*.

The results indicated that the CKD group exhibited a lower number of unique ASVs, a distinct separation from other groups in microbiota composition, and an imbalance between bacteria of the phyla *Firmicutes* and *Bacteroidota*. Additionally, the CKD group demonstrated increased relative frequencies of *Lactobacillus* and *Akkermansia* compared to the control group. These findings suggest significant alterations in gut microbiota in the CKD group. Treatment with live and HK Ln.m effectively mitigated these changes, preventing the uremia-induced disruptions in gut microbiota composition.

## Effects of live and HK Ln.m on colonic uremic toxin concentrations in CKD mice

Uremic toxins derived from gut microbiota, including indole and *p*-Cresol, were elevated in CKD mice. The indole level in the CKD group (65.50 ± 2.11 μg) was significantly higher than in the control group (7.36 ± 4.27 μg). However, treatment with LLp, HKLp, LLn.m, and HKLn.m significantly reduced indole levels in CKD mice ($p < 0.001$) (Fig 4A). Similarly, the

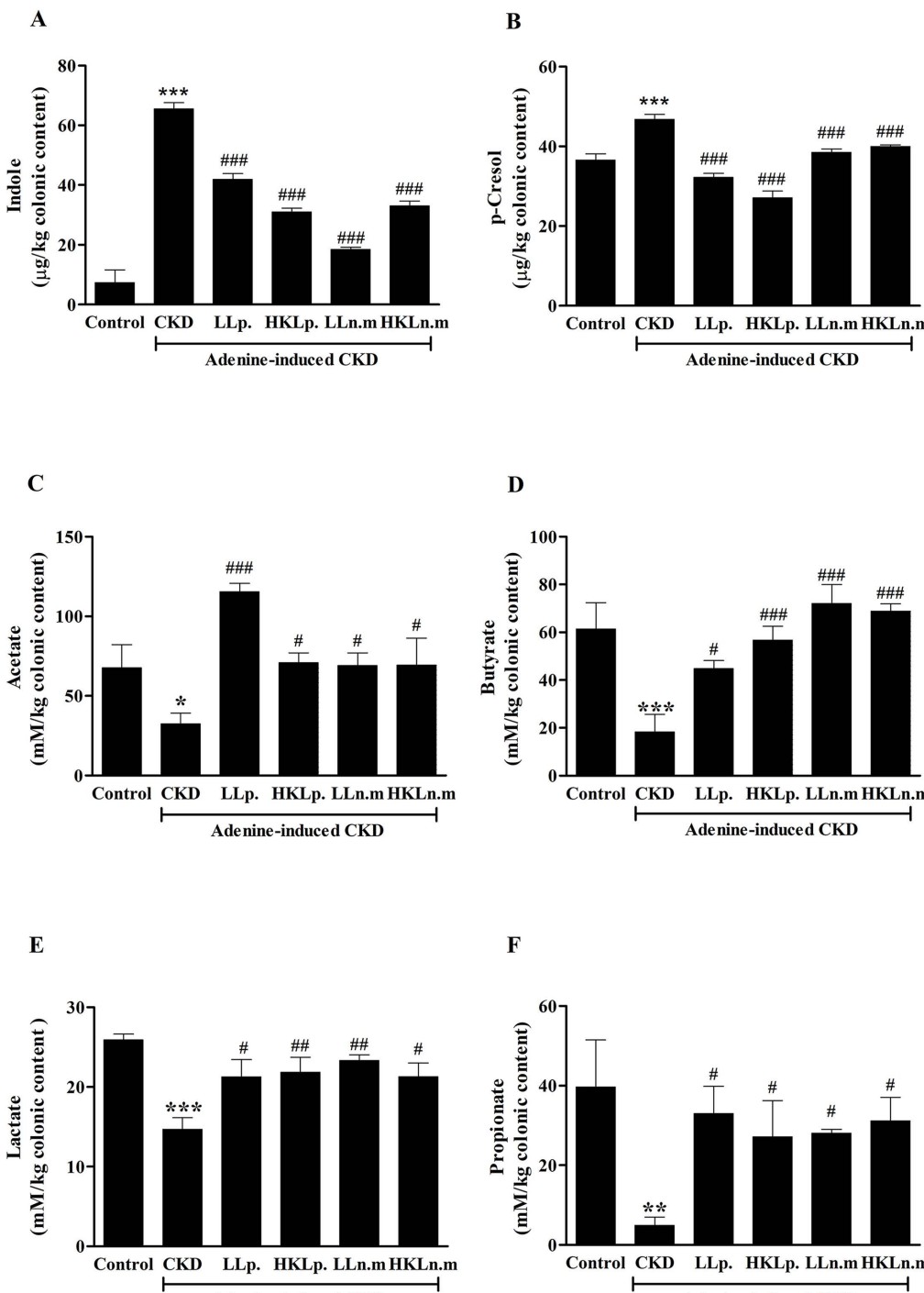

**Fig 4. Effects of live and heat-killed *Leuconostoc mesenteroides* on colonic uremic toxin and short-chain fatty acids (SCFAs) concentrations in chronic kidney disease (CKD) mice.** A: The colonic concentrations of indole. B: The colonic concentrations of p-Cresol. C: The colonic concentrations of acetate. D: The colonic concentrations of butyrate. E: The colonic concentrations of lactate. F: The colonic concentrations of propionate. CKD; chronic kidney disease, LLp; live *Lactiplantibacillus plantarum*, HKLp; Heat-killed *Lactiplantibacillus plantarum*, LLn.m; Live *Leuconostoc mesenteroides*, and HKLn.m; *Leuconostoc mesenteroides*. Data were shown as means ± SEM (n = 5–9). $*p < 0.05$, $**p < 0.01$, and $***p < 0.001$ when compared to control group; $\#p < 0.05$, $\#\#p < 0.01$, and $\#\#\#p < 0.001$ when compared to the CKD group.

*p*-Cresol level was significantly elevated in the CKD group (46.75 ± 1.29 μg) compared to the control group (36.57 ± 1.53 μg). Treatment with LLp, HKLp, LLn.m, and HKLn.m significantly decreased *p*-Cresol levels compared to the untreated CKD group ($p < 0.001$) (Fig 4B). The elevation of these uremic toxin precursors in CKD mice, and their reduction following treatment with live and HK *Ln.m*, likely resulted from gut microbiota modulation.

## Effects of live and HK Ln.m on short-chain fatty acid concentrations in CKD mice

The alteration of gut microbiota in CKD mice may have influenced the levels of SCFAs. A significant decrease in acetate levels was observed in the CKD group (32.39 ± 6.80 mM) compared to the control group (67.59 ± 14.62 mM) ($p < 0.05$). Acetate levels were significantly higher in CKD mice treated with LLp, HKLp, LLn.m, and HKLn.m ($p < 0.001$, $p < 0.05$, $p < 0.05$, and $p < 0.05$, respectively) (Fig 4C). Similarly, butyrate levels were significantly lower in the CKD group (18.28 ± 7.33 mM) compared to the control group (61.36 ± 11.00 mM) ($p < 0.05$). Treatment with LLp, HKLp, LLn.m, and HKLn.m significantly increased butyrate levels compared to the CKD group ($p < 0.05$, $p < 0.01$, $p < 0.001$, and $p < 0.001$, respectively) (Fig 4D). Lactate concentrations were significantly reduced in the CKD group (14.69 ± 1.46 mM) compared to the control group (25.89 ± 0.74 mM) ($p < 0.001$). However, lactate levels significantly increased in CKD mice treated with LLp, HKLp, LLn.m, and HKLn.m compared to the CKD group ($p < 0.05$, $p < 0.01$, $p < 0.01$, and $p < 0.05$, respectively) (Fig 4E). Similarly, propionate levels were significantly lower in the CKD group (4.94 ± 2.02 mM) than in the control group (39.60 ± 11.89 mM) ($p < 0.01$). Treatment with LLp, HKLp, LLn.m, and HKLn.m significantly increased propionate levels compared to the CKD group ($p < 0.05$) (Fig 4F).

These findings suggest that the reduction in SCFAs observed in CKD mice resulted from gut microbiota alterations. Treatment with live and HK Ln.m not only improved gut microbiota composition but also simultaneously enhanced the production of beneficial metabolites.

## Effects of live and HK Ln.m on constipation and intestinal motility in CKD mice

The frequency of defecation was significantly lower in the CKD group (7.3 ± 0.51 times/h) compared to the control group (11.30 ± 1.13 times/h) ($p < 0.05$). CKD groups treated with probiotics, including LLp, HKLp, LLn.m, and HKLn.m, showed a significant increase in defecation frequency compared to the CKD group ($p < 0.01$, $p < 0.01$, $p < 0.05$, and $p < 0.05$, respectively) (Fig 5A). A similar trend was observed in fecal water content. The CKD group exhibited a significant decrease in fecal water content (0.34 ± 0.05%) compared to the control group (0.44 ± 0.01%) ($p < 0.05$). Treatment with LLp, HKLp, LLn.m, and HKLn.m significantly increased fecal water content compared to the CKD group ($p < 0.05$, $p < 0.01$, $p < 0.05$, and $p < 0.01$, respectively) (Fig 5B).

Small intestinal transit was significantly reduced in the CKD group (50.42 ± 2.42%) compared to the control group (60.07 ± 1.97%) ($p < 0.05$). Probiotic treatments with LLp, HKLp, LLn.m, and HKLn.m significantly improved small intestinal transit compared to the CKD group ($p < 0.01$, $p < 0.05$, $p < 0.01$, and $p < 0.05$, respectively) (Fig 5C). Small intestinal smooth muscle motility was significantly lower in the CKD group (1.22 ± 0.60 AU) compared to the control group (2.41 ± 0.30 AU) ($p < 0.01$). Treatment with LLp, HKLp, LLn.m, and HKLn.m significantly improved intestinal motility compared to the CKD group ($p < 0.01$, $p < 0.05$, $p < 0.01$, and $p < 0.05$, respectively) (Fig 5D). Similarly, colonic smooth muscle motility was significantly reduced in the CKD group (0.76 ± 0.27 AU) compared to the control group (1.92 ± 0.28 AU) ($p < 0.01$). However, this loss of motility was significantly reversed in CKD mice

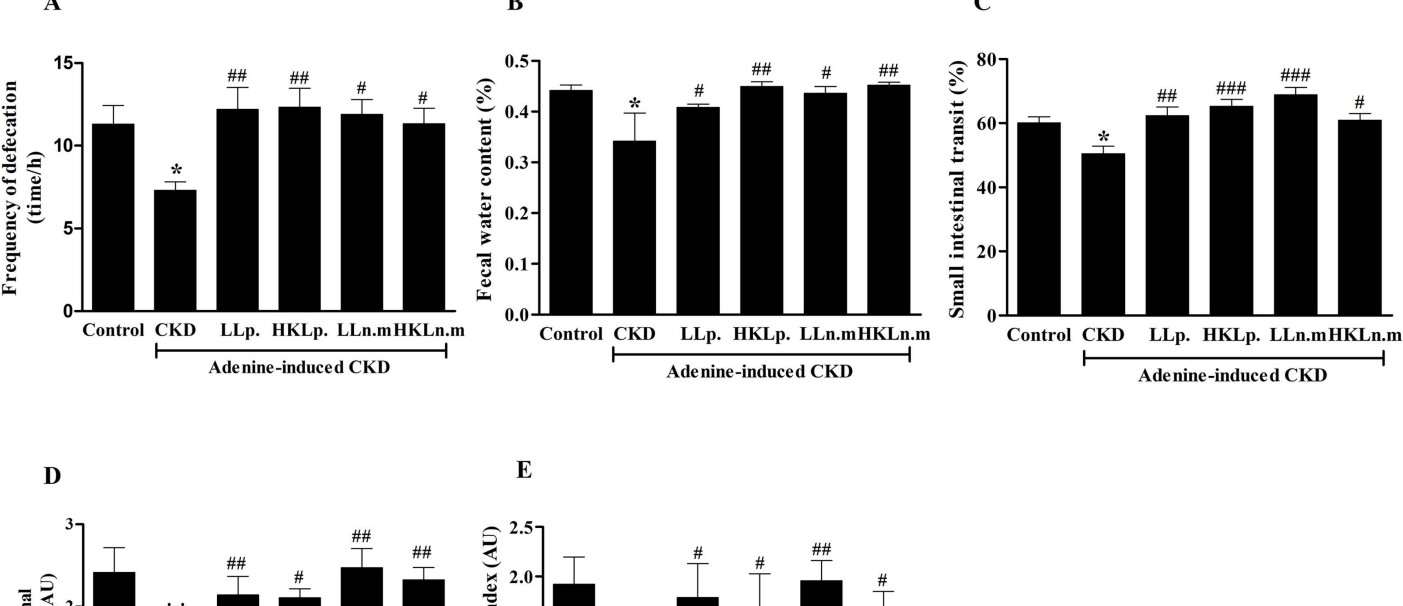

**Fig 5. Effects of live and heat-killed *Leuconostoc mesenteroides* on constipation and intestinal motility in chronic kidney disease (CKD) mice.** A: Frequency of defecation. B: fecal water content. C: small intestinal transit. D: small intestinal motility. E: Large intestinal motility. CKD; chronic kidney disease, LLp; live *Lactiplantibacillus plantarum*, HKLp; Heat-killed *Lactiplantibacillus plantarum*, LLn.m; Live *Leuconostoc mesenteroides*, and HKLn.m; *Leuconostoc mesenteroides*. Data were shown as means ± SEM (n = 5–10). *$p < 0.05$ and **$p < 0.01$ when compared to control group; #$p < 0.05$, ##$p < 0.01$ and ###$p < 0.001$ when compared to CKD group.

treated with LLp, HKLp, LLn.m, and HKLn.m ($p < 0.05$, $p < 0.05$, $p < 0.01$, and $p < 0.05$, respectively) (Fig 5E).

The results indicated that the CKD group experienced a reduction in defecation frequency, fecal water content, and upper gut transit, all of which are indicative of constipation. Additionally, intestinal motility was impaired in CKD mice. However, treatment with both live and HK Ln.m effectively alleviated constipation symptoms in CKD mice.

## Effects of live and HK Ln.m on kidney and intestinal histology in CKD mice

H&E staining revealed abnormal renal corpuscle structure in the kidney tissue of CKD mice. Large spaces were observed between Bowman's capsule and the glomerulus (Fig 6A, marked with a red asterisk), indicating glomerular necrosis in CKD mice. The spaces in the CKD group (10.70 ± 1.00 μm) were significantly larger than those in the control group (4.25 ± 0.30 μm) ($p < 0.001$). However, the spaces were significantly reduced in CKD mice treated with LLp, HKLp, LLn.m, and HKLn.m compared to the CKD group ($p < 0.001$) (Fig 6B). Additionally, tubular dilatation, marked by the green hash symbol in Fig 6A, was evident in the kidneys of CKD mice. The diameter of the distal tubule in the CKD group (18.80 ± 0.21

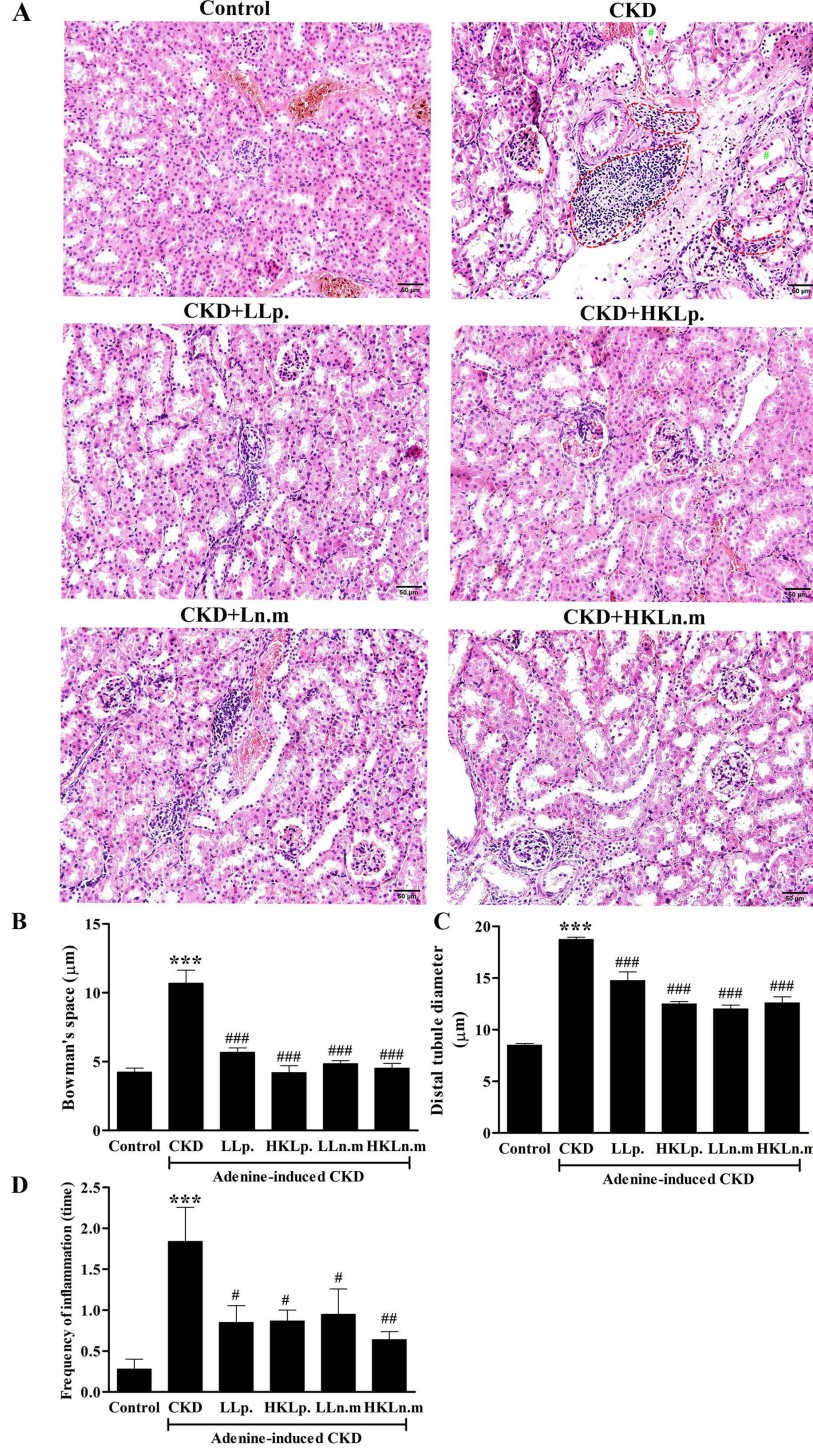

**Fig 6. Effects of live and heat-killed *Leuconostoc mesenteroides* on kidney histopathology in chronic kidney disease (CKD) mice.** A: H & E staining of kidney tissues. Red asterisk: glomerular necrosis; green hash symbol: tubular dilatation; dash line: inflammatory cellular infiltration. B: Glomerular necrosis was determined as the Bowman's space. C: The diameter of distal tubules. D: The frequency of inflammation. CKD; chronic kidney disease, LLp; live *Lactiplantibacillus plantarum*, HKLp; Heat-killed *Lactiplantibacillus plantarum*, LLn.m; Live *Leuconostoc mesenteroides*, and HKLn.m; *Leuconostoc mesenteroides*. Data were shown as means ± SEM (n = 3–5). ***$p < 0.01$ when compared to control group; #$p < 0.05$, ##$p < 0.01$, and ###$p < 0.001$ when compared to the CKD group.

μm) was significantly larger than that in the control group (8.51 ± 0.14 μm) ($p < 0.001$). The tubule diameter was significantly smaller in CKD mice treated with LL*p*, HKLp, LLn.m, and HKLn.m compared to the CKD group ($p < 0.001$) (Fig 6C). Furthermore, cellular infiltration, indicated by the dashed line in Fig 6A, was observed in the CKD kidneys. The frequency of inflammation was significantly higher in the CKD grou*p* (1.84 ± 0.41 times) compared to the control group (0.30 ± 0.12 times) ($p < 0.001$) but was significantly lower in CKD mice treated with LLp, HKLp, LLn.m, and HKLn.m ($p < 0.05$, $p < 0.05$, $p < 0.05$, and $p < 0.01$, res*p*ectively) (Fig 6D).

Masson's Trichrome staining indicated an area of fibrosis in the CKD group (Fig 7A). The area of fibrosis in the CKD group (5.70 ± 0.30%) was significantly larger compared to the control group (3.00 ± 0.23%) ($p < 0.001$). However, the fibrosis area was significantly reduced in CKD mice treated with LLp, HKLp, LLn.m, and HKLn.m compared to the CKD group ($p < 0.01$, $p < 0.001$, $p < 0.001$, and $p < 0.001$, respectively) (Fig 7B).

H&E staining of small intestinal tissue showed damage to the intestinal mucosal layer in CKD mice, including inflammatory cell infiltration (Fig 8A, red dashed line) and villus shortening. Treatment with probiotics alleviated the inflammation, and villus length, measured from the top of the villus to the crypt (indicated by the red line in Fig 8A), was significantly shortened in the CKD group (115.9 ± 9.73 μm) compared to the control group (160.60 ± 6.16 μm) ($p < 0.01$). The villi were significantly longer in the CKD grou*p*s treated with LLp, HKLp, LLn.m, and HKLn.m compared to the CKD group ($p < 0.01$) (Fig 8B). Similarly, the colon showed mucosal damage (red asterisk in S2 Fig.) and inflammation (red dashed line) in CKD mice, which was alleviated by probiotic treatment.

The kidney and intestine of CKD mice displayed significant pathological changes, including glomerular necrosis, tubular dilatation, inflammatory infiltration, and kidney fibrosis. This study highlighted the association between CKD and the gastrointestinal tract, where kidney damage in CKD adversely affected intestinal health. Probiotic treatment with live and HK Ln.m alleviated both kidney and intestinal damage in CKD mice.

## Discussion

Constipation is a common complication among CKD patients. While in healthy individuals, constipation is often caused by various factors and is not typically life-threatening [6], its implication in CKD is more severe. Declining kidney function leads to the accumulation of uremic toxins in the bloodstream, which subsequently enter the gastrointestinal tract, altering the gut microbiota and exacerbating constipation [1]. This creates a cycle where worsening kidney function and increased urea accumulation are linked to more severe constipation. Beyond its impact on quality of life, constipation plays a significant role in CKD progression and is often associated with cardiovascular complications [6,8]. Probiotics have shown promise in modulating gut microbiota, improving gastrointestinal health, and potentially slowing CKD progression [12]. Inactivated probiotics, or paraprobiotics, have emerged as a promising therapeutic option for CKD. These non-viable probiotics offer similar benefits to live probiotics while reducing the risk of microbial translocation, particularly in vulnerable populations. However, further research is needed to fully understand their effects in CKD contexts [3]. This study aimed to evaluate the effects of both live and heat-killed (HK) *Leuconostoc mesenteroides* (Ln.m) probiotics on gastrointestinal functions in CKD mice.

The ability of lactic acid bacteria (LAB) strains to survive in gastric juice and bile salt is critical for their viability and functionality as probiotics. To benefit the host, LAB strains must withstand highly acidic environments (pH 2.5 to 3.5) and bile salt concentrations ranging from 0.03% to 0.3% [31]. Among the strains studied, strain P.6.6 demonstrated the highest

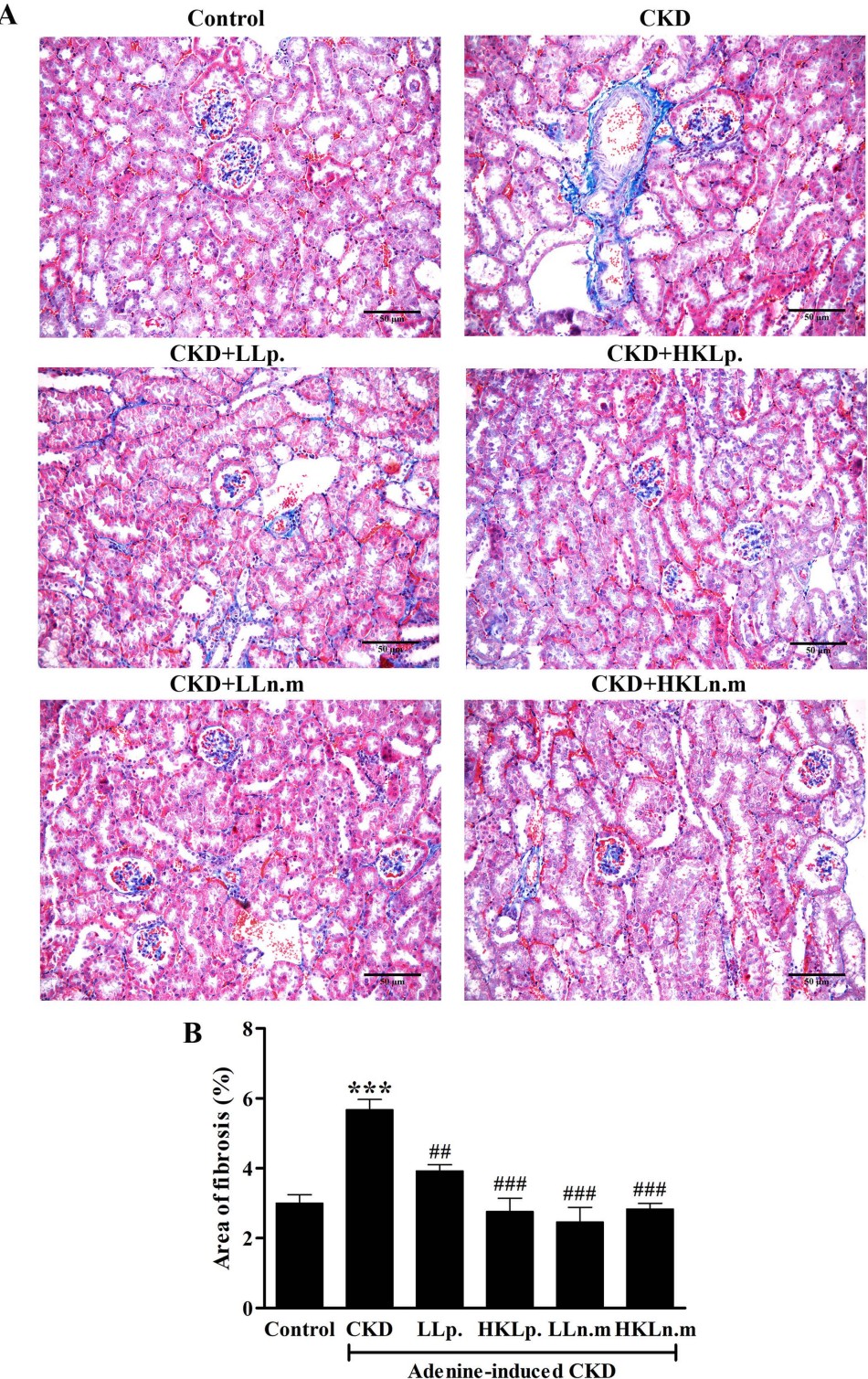

**Fig 7. Effects of live and heat-killed *Leuconostoc mesenteroides* on renal fibrosis.** A: Masson's Trichome staining of kidney tissue. B: Area of fibrosis (%). CKD; chronic kidney disease, LLp; live *Lactiplantibacillus plantarum*, HKLp; Heat-killed *Lactiplantibacillus plantarum*, LLn.m; Live *Leuconostoc mesenteroides*, and HKLn.m; *Leuconostoc mesenteroides*. Data were shown as means ± SEM (n = 3–5). ***$p < 0.01$ when compared to control group; ##$p < 0.01$ and ###$p < 0.001$ when compared to CKD group.

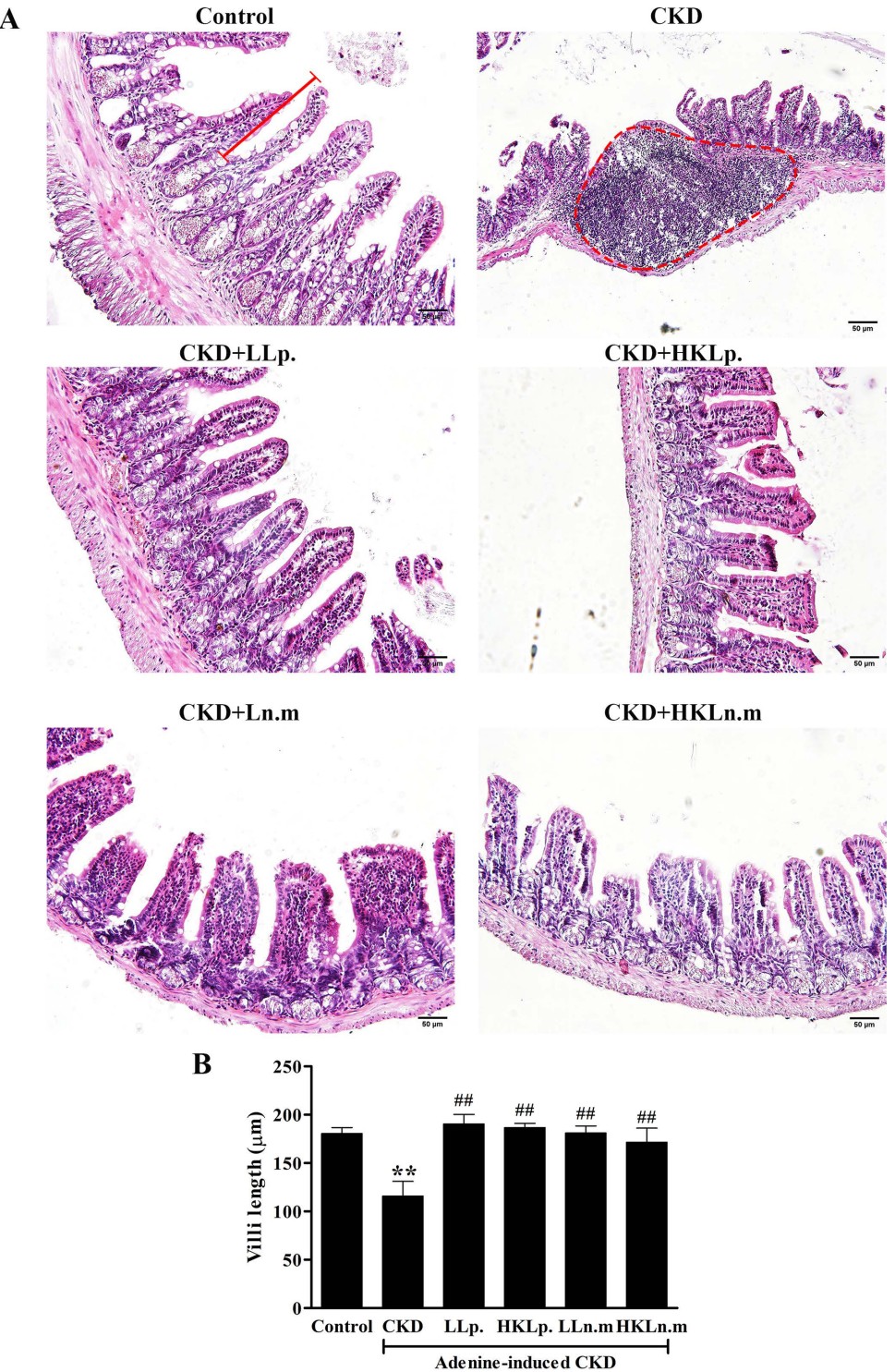

**Fig 8. Effects of live and heat-killed *Leuconostoc mesenteroides* on small intestinal histology.** A: H & E staining of small intestinal tissues. Red dash line: inflammatory cell infiltration; red line: the length of villi which was measured from the top of the villus to the crypt. B: The measurement of villi length. CKD; chronic kidney disease, LLp; live *Lactiplantibacillus plantarum*, HKLp; Heat-killed *Lactiplantibacillus plantarum*, LLn.m; Live *Leuconostoc mesenteroides*, and HKLn.m; *Leuconostoc mesenteroides*. Data were shown as means ± SEM (n = 3). **$p < 0.01$ when compared to control group; ##$p < 0.01$ when compared to CKD group.

survival rate under these conditions. Previous studies have shown that bacteria capable of tolerating acid and bile salts without significant cell loss possess probiotic properties and exhibit strong intestinal viability [15,31]. Antibiotic susceptibility testing was conducted to assess the safety of the LAB strains, given that plasmid-encoded antibiotic resistance genes can transfer between bacterial strains, posing a potential health risk [32]. Susceptibility was evaluated using 13 different antibiotic discs [23]. Most LAB strains were sensitive to all antibiotics tested, except vancomycin. Previous research has indicated that LAB isolated from natural sources are resistant to β-lactam antibiotics, such as ampicillin, but sensitive to principal antibiotics like chloramphenicol, erythromycin, and tetracycline [33]. LAB strains from fermented foods have also shown resistance to antibiotics primarily targeting gram-negative bacteria, such as amikacin, aztreonam, ciprofloxacin, gentamycin, kanamycin, streptomycin, sulfamethoxazole, and vancomycin. Importantly, this resistance is an intrinsic property of the LAB genus and is not transferable to pathogenic bacteria or microbial flora, ensuring their safety as probiotics [20]. The acid and bile salt tolerance studies confirmed that strain P.6.6, identified as *Leuconostoc mesenteroides*, could survive in the gastrointestinal tract. In addition, hemolytic activity and antibiotic susceptibility demonstrated the strain's safety for probiotic use. Consequently, strain P.6.6 was selected for use as a probiotic in an adenine-induced CKD study, where its probiotic properties were compared with those of *Lactiplantibacillus plantarum*, a standard probiotic.

*L. mesenteroides* (Ln.m) is a gram-positive, heterofermentative lactic acid bacterium (LAB) renowned for its health-promoting properties as a probiotic. It is commonly found in fermented foods, beverages, and in the gastrointestinal tracts of fish and shrimp [34,35]. Research has highlighted the probiotic potential of Ln.m in various contexts. For instance, Ln.m isolated from kimchi demonstrated its ability to prevent lead accumulation in mice by enhancing lead resistance and promoting its removal [36]. Another strain, Ln.m (FB111), isolated from mustard kimchi, showed potential in preventing cholesterol absorption in Caco-2 cells, suggesting its beneficial role in the food industry [37]. Moreover, Ln.m (EH-1), isolated from Mongolian curd cheese, was shown to reduce blood glucose levels in type-1 diabetic mice by increasing butyric acid concentrations. The effect is mediated through the free fatty acid receptor 2 (Ffar2) [35]. Beyond live probiotics, inactivated forms of Ln.m or its exopolysaccharides also offer potential health benefits. For example, dextran produced by Ln.m (742) was found to enhance the growth of *Bacteroides* and other beneficial bacteria in an in vitro human fecal fermentation model, indicating its potential to support gut health [38].

In this study, we assessed the effects of live and heat-killed (HK) *L. mesenteroides* (Ln.m) on CKD mice. From the onset of adenine-induced CKD to the beginning of the treatment period, water intake in the CKD group increased, while body weight (BW) decreased. These observations indicated the onset of renal failure in the early stages of CKD in the adenine-induced model. The high volume of water excretion observed is an early effect of adenine-induced CKD, where adenine downregulates sodium-potassium chloride cotransporters (NKCC) and aquaporin 2 (AQP2) in renal epithelial cells, leading to significant fluid loss and increased urine output. In addition to fluid loss, impaired kidney function resulted in elevated BUN and plasma creatinine levels, consistent with findings from previous studies [39,40]. Treatment with live and HKLn.m resulted in decreased BUN and creatinine levels in CKD mice. While the specific effects of Ln.m as a probiotic in CKD have not been widely reported, previous studies on probiotics in both human and animal models have documented reductions in BUN and creatinine levels, which helped alleviate CKD symptoms [3,12]. Additionally, the exopolysaccharides produced by Ln.m were shown to alleviate CKD symptoms in mice by improving gastrointestinal function. These compounds reduced lipopolysaccharide-binding protein, endotoxin levels, and intestinal barrier defects in CKD

mice [41]. The improvements in gastrointestinal function were reflected in the decreased plasma levels of BUN and creatinine, highlighting the potential benefits of Ln.m as a therapeutic agent for CKD.

Uremic dysbiosis in CKD is characterized by a shift in the gut microbiota, marked by a reduction in beneficial bacteria and an increase in pathogenic, proteolytic, and urease-producing bacteria. The gut microbiota in CKD exhibits distinct alterations: first, reduced abundance and diversity of gut microbiota, second, decreased *Lactobacillus* population and changes in the *Bacteroides* population, and lastly, increased production of metabolites and toxins, with a decrease in SCFAs. The gut microbiota communicates with the host through the enteric nervous system (ENS) and vagus nerve which control intestinal homeostasis [42,43]. Therefore, the microbial changes impair gastrointestinal functions, including barrier integrity, absorption, and motility [2,3,12,43]. In our study, we observed an imbalance between the Firmicutes and Bacteroidota phyla in CKD, a finding indicative of gut dysbiosis. Interestingly, we also noted an increase in *Lactobacillus* which is a result consistent with the previous finding [5]. A possible explanation for this is that CKD leads to reduced intestinal protein absorption. Some *Lactobacillus* strains, such as *Lactobacillus gasseri*, can utilize purines and reduce purine absorption. Furthermore, CKD-induced reductions in intestinal motility could prolong microbial fermentation or proteolysis, which may favor the proliferation of *Lactobacillus* [44,45]. Our results indicated that both live and heat-killed Ln.m (HKLn.m) were effective in reversing gut microbiota alterations. They increased the population of beneficial gut microbiota, reduced uremic toxin precursors, and boosted SCFAs production in CKD mice. Probiotics modulate the gut microbiota primarily through two mechanisms: i) enhancing epithelial cell proliferation and integrity, promoting inflammatory response, and improving the gastrointestinal mucosa and ii) directly modifying the gut microbiota by improving the gut environment. These actions are bidirectional, as improvements in epithelial integrity affect microbiota composition and vice versa. In contrast, inactivated probiotics, such as HKLn.m, can only modify the gut microbiota by promoting a healthy gastrointestinal mucosa, which, in turn, affects the microbiota composition [46].

Constipation is a common complication in CKD patients, often linked to gastrointestinal dysfunction and alterations in gut microbiota. These changes play a significant role in the progression of CKD to end-stage kidney disease (ESKD) [6,9]. The gut microbiota alterations in CKD are not only associated with the enhancement of harmful metabolites such as indoxyl sulfate, p-cresyl sulfate, hippuric acid, or trimethylamine N-oxide (TMAO) but also with a reduction in beneficial metabolites, like SCFAs [47]. In this study, CKD mice treated with *L. mesenteroides* (Ln.m) exhibited improved defecation status and enhanced intestinal smooth muscle motility. These findings are consistent with previous reports, where inactivated probiotics were shown to improve gastrointestinal symptoms in hemodialysis children. Symptoms such as anorexia, nausea, vomiting, abdominal discomfort, distension, and constipation were significantly reduced after one to two months of treatment [48]. The effects of *L. mesenteroides* on gastrointestinal functions in this study may result from the modulation of the intestinal environment, including the regulation of gut microbiota, reduction of uremic toxin precursors (such as indole and p-Cresol), and an increase in SCFAs levels. The gut microbiota plays a crucial role in the development and maturation of the ENS, which in turn affects gut motility. It was reported that the germ-free condition affects the bacterial composition leading to gastrointestinal dysfunctions [49]. Gastrointestinal transit time was found to be increased in germ-free mice, and colonization by specific pathogens could stimulate the migrating motor complex and normalize gut transit time [50]. Microbial metabolites like lactate and SCFAs (acetate, propionate, and butyrate) enhance gastrointestinal motility through several mechanisms, including interactions with serotonin (5-HT), and the regulation of motor and

secretory functions in the ENS. The ENS itself stimulates secretion, motility, and contractile activity, directly promoting gut motility [48]. Furthermore, colonic infusion with SCFAs has been shown to reduce intestinal transit time and stimulate the mucosal and vagus nerves, further enhancing intestinal smooth muscle contraction [50,51].

In addition to elevated BUN and creatinine levels, kidney structural damage is a significant indicator of CKD progression in animal models, reflecting the pathophysiology observed in humans. Structural changes in the kidney include tubular atrophy, focal hypertrophy, glomerular necrosis, and fibrosis, although the specific changes may vary depending on the animal model used [52]. Our histopathological analysis revealed that the kidneys of CKD mice exhibited glomerular necrosis, tubular dilatation, inflammatory cell infiltration, and fibrosis. The mechanism behind adenine-induced kidney damage involves the metabolism of adenine via the xanthine pathway, producing 2,8-dihydroxyadenine (DHA). DHA is deposited in the renal tubules, leading to renal inflammation, structural damage, and progressive functional decline over time [39]. Additionally, in CKD, indoxyl sulfate (IS), a metabolite derived from gut microbiota processing of tryptophan, accumulates and correlates with CKD progression. IS induces inflammation, increases oxidative stress in the kidneys, and contributes to glomerular sclerosis and renal fibrosis [53]. In this study, we observed increased levels of IS and p-Cresol sulfate (PCS) precursors, which are linked to renal fibrosis in CKD. Histopathological changes were also detected in the small and large intestines of CKD mice. Treatment with *L. mesenteroides* (Ln.m) resulted in the improvement of all pathological damage observed in both the kidneys and intestines. This connection between CKD and gastrointestinal dysfunction has been well-documented in previous studies [3,12,40,54]. A proposed mechanism suggests a vicious cycle between CKD, constipation, and gut microbiota alterations that contribute to CKD progression. Our findings align with previous reports suggesting that the modulation of the gut microbiota and intestinal environment may alleviate CKD progression [55].

## Conclusion

This study demonstrated that *L. mesenteroides* (Ln.m) exhibits probiotic properties and that heat-killed *L. mesenteroides* (HKLn.m) is equally effective as live Ln.m in alleviating constipation and slowing CKD progression in a mouse model. The beneficial effects of both live and HK Ln.m appear to be mediated through modulation of the gut microbiota and an increase in SCFAs production. Based on these findings, we propose that *L. mesenteroides* (Ln.m) can be considered a promising probiotic, with HKLn.m potentially offering a novel therapeutic approach for improving constipation in CKD patients and slowing the progression of the disease. However, further research is needed to explore the effects of HKLn.m on other gastrointestinal functions, such as the intestinal barrier and luminal environment, and to better understand the mechanisms by which HKLn.m influences gastrointestinal health.

## Supporting information

**S1 Fig. Effects of live and HK *Leuconostoc mesenteroides* on liver function in CKD mice.** The liver function was investigated by enzyme levels after the treatment period. A, B, and C: The AST, ALT, and ALP levels, respectively. CKD; chronic kidney disease, LLp; live *Lactiplantibacillus plantarum*, HKLp; Heat-killed *Lactiplantibacillus plantarum*, LLn.m; Live *Leuconostoc mesenteroides*, and HKLn.m; *Leuconostoc mesenteroides*. Data were shown as mean ± SEM (n = 7–10). #$p < 0.05$ when compared to the control group; *$p < 0.05$ and **$p < 0.01$ when compared to the CKD group (one-way ANOVA repeated by Bonferroni test).
(TIF)

**S2 Fig. Effects of live and HK *Leuconostoc mesenteroides* on large intestine histology.** The H & E staining of small intestinal tissue was stained with H & E to investigate the histological morphology of the large intestine in CKD mice. The structure of the colon was the same way with the small intestine, the tissue staining showed mucosal damage and inflammation in the CKD group. While in CKD treated with probiotic groups ameliorated this effect of CKD on the colon. CKD; chronic kidney disease, LLp; live *Lactiplantibacillus plantarum*, HKLp; Heat-killed *Lactiplantibacillus plantarum*, LLn.m; Live *Leuconostoc mesenteroides*, and HKLn.m; *Leuconostoc mesenteroides*.
(TIF)

**S1 Table. Antibiotic susceptibility of the isolated lactic acid bacteria.**
(TIF)

**S2 Table. Effects of heat-inactivated *Leuconostoc mesenteroides* on hematological parameters.**
(PDF)

**S3 Table. Nutritional composition of mice feed.**
(PDF)

**S1 File. Raw data.**
(XLSX)

**S2 File. Raw data for ASV analysis.**
(XLSX)

## Acknowledgment

The authors are grateful to Mr. Thomas Coyne, Faculty of Science, Prince of Songkla University for assisting in proofreading and providing feedback on the manuscript.

## Author contributions

**Conceptualization:** Fittree Hayeeawaema, Natthawan Sermwittayawong, Nawiya Huipao, Paradorn Muangnil, Pissared Khuituan.

**Data curation:** Fittree Hayeeawaema, Chittipong Tipbunjong, Pissared Khuituan.

**Formal analysis:** Fittree Hayeeawaema, Natthawan Sermwittayawong, Chittipong Tipbunjong, Pissared Khuituan.

**Funding acquisition:** Fittree Hayeeawaema.

**Investigation:** Fittree Hayeeawaema, Natthawan Sermwittayawong, Chittipong Tipbunjong, Nawiya Huipao, Paradorn Muangnil, Pissared Khuituan.

**Methodology:** Fittree Hayeeawaema, Natthawan Sermwittayawong, Chittipong Tipbunjong, Nawiya Huipao, Paradorn Muangnil, Pissared Khuituan.

**Project administration:** Natthawan Sermwittayawong, Pissared Khuituan.

**Resources:** Natthawan Sermwittayawong.

**Supervision:** Natthawan Sermwittayawong, Nawiya Huipao, Pissared Khuituan.

**Validation:** Fittree Hayeeawaema, Natthawan Sermwittayawong, Chittipong Tipbunjong, Paradorn Muangnil, Pissared Khuituan.

**Visualization:** Fittree Hayeeawaema, Natthawan Sermwittayawong, Chittipong Tipbunjong, Pissared Khuituan.

**Writing – original draft:** Fittree Hayeeawaema, Pissared Khuituan.

**Writing – review & editing:** Fittree Hayeeawaema, Natthawan Sermwittayawong, Chittipong Tipbunjong, Nawiya Huipao, Paradorn Muangnil, Pissared Khuituan.

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
