## [Decision Letter · Decision Letter 0]

3 Dec 2024

PONE-D-24-41563Live and heat-killed Leuconostoc mesenteroides counteract the gastrointestinal dysfunction in chronic kidney disease mice through intestinal environment modulationPLOS ONE

Dear Dr. KHUITUAN,

Thank you for submitting your manuscript to PLOS ONE. After careful consideration, we feel that it has merit but does not fully meet PLOS ONE’s publication criteria as it currently stands. Therefore, we invite you to submit a revised version of the manuscript that addresses the points raised during the review process. Please consider the suggestions done by the reviewers to improve the manuscript

We look forward to receiving your revised manuscript.

Kind regards,

Guadalupe Virginia Nevárez-Moorillón, Ph.D.

Academic Editor

PLOS ONE

“FH: the National Research Council of Thailand (NRCT) (NRCT5-RGJ63-160)”

Reviewers' comments:

Reviewer's Responses to Questions

**Comments to the Author**

1. Is the manuscript technically sound, and do the data support the conclusions?

Reviewer #1: Partly

Reviewer #2: Yes

Reviewer #3: Yes

2. Has the statistical analysis been performed appropriately and rigorously? 

Reviewer #1: No

Reviewer #2: Yes

Reviewer #3: Yes

3. Have the authors made all data underlying the findings in their manuscript fully available?

Reviewer #1: No

Reviewer #2: Yes

Reviewer #3: Yes

4. Is the manuscript presented in an intelligible fashion and written in standard English?

Reviewer #1: No

Reviewer #2: Yes

Reviewer #3: Yes

5. Review Comments to the Author

Reviewer #1: This study aims to evaluate the probiotic potential of lactic acid bacteria from Tiger prawn and to investigate the effects of selected probiotics both live and heat-killed on renal and gastrointestinal functions in CKD mice. The study presents the results of primary scientific research. Nevertheless, the manuscript and experiments still showed many problems.

1. Why authors choose to isolated LAB strains from Tiger prawn?

2. In line 105, “the gastrointestinal tract” should be changed to “the content of gastrointestinal tract”

3. Why isolated LAB strains were cultured with shaking at 150 rpm?

4. The procedure of strains isolation is not detailed enough. For example, when the incubated broth of isolated sample was plated on the MRS agar, had the broth undergone gradient dilution?

5. In line 108, the “24 h.” should be modified as “24 h”.

6. In line 111, the “Lactobacillus plantarum ATCC 14917 (TISTR 877) and Lactobacillus casei ATCC 7469” should be changed to “Lactiplantibacillus plantarum ATCC 14917 (TISTR 877) and Lacticaseibacillus casei ATCC 7469”.

7. The methods and results for isolated strain identification were missing.

8. Why Lactiplantibacillus plantarum ATCC 14917 (TISTR 877) and Lacticaseibacillus casei ATCC 7469 were used as probiotic standards?

9. In line 154-155, the strain information of L. plantarum is missing.

10. In line 156-157, the strain information of Leuconostoc mesenteroides is missing.

11. The numbers of mice in each experimental group should be given.

12. In Fig 3B, all the samples of each group should be illustrated.

13. Why not evaluated the alpha-diversity of gut flora?

Reviewer #2: The study design is appropriate. The results have been properly reported and discussed by using relevant literature. As specific comments, please check for typos in the full text. Also, make sure that all of the references have been reported in the text or in the references list. It should be noted that the authors indicated the composition of the ration that the mice received. 

Reviewer #3: I am writing to submit my review report for the manuscript entitled “Live and heat-killed Leuconostoc mesenteroides counteract the gastrointestinal dysfunction in chronic kidney disease mice through intestinal environment modulation " for your consideration. Overall, I find the manuscript's findings intriguing and the information provided useful for researchers and academia. The article has the potential to make a significant contribution to the related discipline..

However, I have some concerns regarding the clarity, detail, and precision of different sections, which I outline below:

I recommend that the authors address these concerns and provide a revised version of the manuscript for further consideration

• Abstract-Keep a proper sequence in the abstract and end with conclusion. Focus on the gap you have covered in your study

• L-23 Abstarct- mention numerical values for better understanding>>CKD mice 23 treated with live and heat-killed Ln.m showed blood urea nitrogen and creatinine levels significantly 24 increased in the CKD compared to the control group, nevertheless, they were significantly reduced 25 in both live and heat-killed probiotic-treated groups. Kidney damage, Firmicutes/Bacteroidota 26 imbalance, increasing colonic uremic toxin, decreasing fecal short-chain fatty acids, and constipation 27 were observed in CKD.

• L-29 Abstract Reconsider and remove repetition - Taken together, Ln.m could be considered 29 a probiotic, and heat-killed Ln.m exhibits a similar effect to its live form in alleviating gastrointestinal 30 dysfunction and the progression of renal damage in CKD mic

• L-59 Introduction – is it a reference statement???????????? As a result, there is 58 growing interest in strategies to manipulate and restore beneficial microbiota to improve health outcomes in the female 59 reproductive tra

• L-47 Need little attention for better understanding - Notably, short-chain 47 fatty acids (SCFAs), such as acetic, propionic, and butyric acids, which are microbial metabolites, 48 decreased along with the decreasing diversity of gut microbiota in CKD mice (5). It was reported that 49 the alteration of gut microbiota and consequent reduction of SCFAs production played

• L-58-check the cited reference- Prolonged intestinal transit has bidirectional effects with gut microbiota alteration that 59 result in increased gut-derived uremic toxins such as indole and p-Cresol which will be absorbed and 60 metabolized into indoxyl sulfate and p-Cresol sulfate, respectively. Indoxyl sulfate and p-Cresol 61 sulfate are normally excreted by the kidneys, but in CKD, the toxins accumulate in the circulation 62 and cause kidney fibrosis (6–8)

• L-92 You mean parabiotics???????????? ). Using inactivated probiotics could avoid the potential risks of live probiotics on 92 vulnerable or pediatric patients by translocating from the gastrointestinal tract to the blood of these 93 patients. In addition to the safety aspect, inactivated probiotics are easier to transport, store, and 94 standardize than live probiotics (‘

• The introduction could be improved by providing more context and background from following latest references,

o doi: https://doi.org/10.1016/j.jep.2023.116503

o doi: https://doi.org/10.1016/j.ejphar.2024.176356

o https://doi.org/10.1016/j.lfs.2023.122380

o .doi: 10.3389/fnut.2024.1364841

• L-105 Please elaborate the conditions for better understanding- To isolate the LAB from the gastrointestinal tract of Tiger prawn, the gastrointestinal tract 106 was collected under hygienic conditions. The sample was homogenized and incubated in the De Man, 107 Rogosa, and Sharpe (MRS) broth at 37℃ with shaking at 150 rpm for 24 h. It was plated on the MRS 108 agar and incubated at 37℃ for 24 h. in the

• L-150 Check RPM- aerobic condition at 37ºC with shaking at 150 rpm for 24 h. The bacteria were harvested 143 by centrifugation at 8,000 rpm for 10 min and washed twice with 0. 85% NaCl. Bacterial cells were 144 resuspended in 0. 85% NaCl and then inactivated in the water bath at 100ºC for 10 min. The HK 145 bacteria were freeze- dried and kept at –80ºC until used. The freeze- dried bacteria powder was 146 suspended in sterile distilled water to be administer

• L-159 need clarity ----After 28 days of induction, mice were administered treatments once a day for 28 days. 160 Freeze-dried live and HK probiotics were re-suspended in sterilized distilled water freshly before oral 161 gavage with a final volume of 0.2 mL/mouse/day. Body weight (BW), and food and water intake of 162 all mice were measured every day during the experimental period and their general health was 163 monitored. At the end of the experiment,

• This statement is creating confusion- Another part of the whole blood was centrifuged at 4,000 rpm for 10 min. From this sample, 175 BUN, plasma creatinine, aspartate transaminase (AST), alanine transaminase (ALT), and alkaline 176 phosphatase (ALP) levels were measured by BS-20 Chemistry Analyzer (Mindray, Shenzhen, 177 China

• Cite the following latest references in discussion section

o doi: 10.3389/fphar.2022.897926

o doi: 10.2147/DDDT.S107917

o doi: 10.3389/fphar.2018.01461

o 10.1186/s13020-023-00745-5

o doi: 10.3389/fphar.2022.1022567

o doi: https://doi.org/10.3389/fphar.2023.1166022

o

• L-212Insert as equation- he distance of the charcoal marker traveled and the total small intestinal length was 211 measured with cotton thread immediately after anesthetization and abdominal incisions were made. 212 Gastrointestinal transit (%) was calculated as (distance of charcoal meal marker/total length of the 213 small intestine) × 100.

• What about other metabolites- The alteration of gut microbiota in CKD was not only related to the enhancement of harmful 510 metabolites (uremic toxins) but also associated with the reduction of beneficial metabolites (SCFAs) 511 (45). In the present study, CKD mice treated with Ln.m showed improved defecation status and 512 intestinal smooth muscle motility

• Tables check the duplications-able

• Italic all the scientific names,

• Remove grammatical mistakes

• Need to rewrite the conclusion

Recheck Legends description is as per figure number and discussion-

I urge the authors to improve the English language for better flow of literature.

Please check reference style throughout MS

6. PLOS authors have the option to publish the peer review history of their article (what does this mean? ). If published, this will include your full peer review and any attached files.

**Do you want your identity to be public for this peer review?** For information about this choice, including consent withdrawal, please see our Privacy Policy .

Reviewer #1: No

Reviewer #2: No

Reviewer #3: **Yes: ** Dr. Muhammad Afzaal

---

## [Author Response · Author response to Decision Letter 0]

16 Jan 2025

Manuscript Ref. No.: PONE-D-24-41563

Title: Live and heat-killed Leuconostoc mesenteroides counteract the gastrointestinal dysfunction in chronic kidney disease mice through intestinal environment modulation

Fittree Hayeeawaema, Natthawan Sermwittayawong, Chittipong Tipbunjong, Nawiya Huipao, Paradorn Muangnil, Pissared Khuituan

Responses to Comments by Editor and Reviewers

Editor's and Reviewers' comments:

Editor:

We are very grateful to the Reviewing Editor for his interest in our work and inviting us to submit a revised manuscript. We have carefully considered the reviewers’ recommendations and comments. As the Reviewing Editor asked about meeting PLOS ONE style templates, we have edited the manuscript following the guidelines.

2. Thank you for stating the following financial disclosure: “FH: the National Research Council of Thailand (NRCT) (NRCT5-RGJ63-160)” Please state what role the funders took in the study. If the funders had no role, please state: "The funders had no role in study design, data collection and analysis, decision to publish, or preparation of the manuscript." If this statement is not correct you must amend it as needed. Please include this amended Role of Funder statement in your cover letter; we will change the online submission form on your behalf.

The funder had no role in this study and we have stated this role in the manuscript.

We (all authors) decided to share the raw data in the supplemental file.

We have considered the editor's recommendation and decided to remove the phrase “data not shown” which is not a core part of the research.

Reviewer #1:

This study aims to evaluate the probiotic potential of lactic acid bacteria from Tiger prawn and to investigate the effects of selected probiotics both live and heat-killed on renal and gastrointestinal functions in CKD mice. The study presents the results of primary scientific research. Nevertheless, the manuscript and experiments still showed many problems.

1. Why authors choose to isolated LAB strains from Tiger prawn?

We thank the reviewer for the careful evaluation of our manuscript. We performed the preliminary study to isolate the LAB from various natural sources including fermented cabbage, fermented garlic, the gastrointestinal content of Sea bass, and the gastrointestinal content of the Tiger prawn. Next, we screened the LAB from different sources by acid tolerance tests and we found that only the LAB from Tiger prawn gastrointestinal content could survive after one hour. We have added the detail in the manuscript (see Page 7, lines 113-114, 118-120)

2. In line 105, “the gastrointestinal tract” should be changed to “the content of gastrointestinal tract”

Following the recommendation of the reviewer, the context was changed. (see Page 7, line 114)

3. Why isolated LAB strains were cultured with shaking at 150 rpm?

The shaking condition was according to a previous study (Soundharrajan et al., 2023). The lactic acid bacteria are mostly facultative anaerobic. Practically, shaking 150 RMP was to mix the nutrition in the media through the tube and avoid the oxygen gradient.

4. The procedure of strains isolation is not detailed enough. For example, when the incubated broth of isolated sample was plated on the MRS agar, had the broth undergone gradient dilution?

Following the reviewer's recommendation, we have added more details following the reviewer’s suggestion. After the samples were homogenized and incubated in MRS broth, the culture media was diluted as serial dilution before spread on the MRS agar. (see Page 7, line 116-117)

5. In line 108, the “24 h.” should be modified as “24 h”.

The abbreviation of an hour was modified as “h” throughout the manuscript.

6. In line 111, the “Lactobacillus plantarum ATCC 14917 (TISTR 877) and Lactobacillus casei ATCC 7469” should be changed to “Lactiplantibacillus plantarum ATCC 14917 (TISTR 877) and Lacticaseibacillus casei ATCC 7469”.

The name of the bacteria was changed throughout the manuscript.

7. The methods and results for isolated strain identification were missing.

MALDI Biotyper identified the isolated bacteria; the report is shown below (from the Scientific Equipment Center).

8. Why Lactiplantibacillus plantarum ATCC 14917 (TISTR 877) and Lacticaseibacillus casei ATCC 7469 were used as probiotic standards?

This study was to evaluate the probiotic properties of lactic acid bacteria from natural sources to use as a probiotic treatment in chronic kidney disease (CKD) mice. The Lactiplantibacillus plantarum and Lacticaseibacillus casei are probiotics that have been found in natural sources and have promising effects in a wide range such as antioxidant, improve constipation, and irritable bowel disease (IBD) (Jang et al., 2018; Swain et al., 2014; Yamada et al., 2018). Moreover, this study also investigated the effect of heat-inactivated probiotics on the gastrointestinal tract of CKD mice and both Lactiplantibacillus plantarum and Lacticaseibacillus casei have been reported that they provide positive effects on the gastrointestinal tract (Jang et al., 2018; Sang et al., 2015). Therefore, using Lactiplantibacillus plantarum and Lacticaseibacillus casei is to expect a similar or better effect from selected LAB.

9. In line 154-155, the strain information of L. plantarum is missing.

The ATCC of L. plantarum has been included. (see Page 9, line 166)

10. In line 156-157, the strain information of Leuconostoc mesenteroides is missing.

We don’t have the ATCC of Leuconostoc mesenteroides isolated from Tiger prawn.

11. The numbers of mice in each experimental group should be given.

We added the number of mice in the manuscript.

12. In Fig 3B, all the samples of each group should be illustrated.

The study aimed to screen the broad trending of microbial shifting on group–level (Ray et al., 2019). Therefore, we used the pooled method which has long been used for bacterial community screening such as infection, antibiotic use, and CKD population (Banjong et al., 2023). Therefore, the sample of each group in Fig 3B was a pooled – DNA from the colonic content of each mouse in the group.

13. Why not evaluated the alpha-diversity of gut flora?

Since alpha diversity of gut flora is the biodiversity within an individual group, in this study, we focus on the changes at the group level. Hence, the pooled sample could not provide the calculation of alpha-diversity in each group.

Reviewer #2: The study design is appropriate. The results have been properly reported and discussed by using relevant literature. As specific comments, please check for typos in the full text. Also, make sure that all of the references have been reported in the text or in the references list. It should be noted that the authors indicated the composition of the ration that the mice received.

We thank the reviewer for the interest in our work and valuable comments on our manuscript. We have checked and edited the manuscript according to the reviewer’s recommendation. For the nutritional composition, we have reported in the supplemental data. It contains 12% moisture, 24% crude protein, 4.5% fat, 5% fiber, 1.0% calcium, 0.9% phosphorus, 0.20% sodium, 1.17% potassium, 0.23% magnesium, 171 p.p.m. manganese, 22 p.p.m. copper, 100 p.p.m. zinc, 180 p.p.m. iron, 1.82 p.p.m. cobalt, 1 p.p.m. potassium iodide, 0.1 p.p.m. selenium, Vitamins; A: 20,000 i.u./kg, D: 4,000 i.u./kg, E: 100 mg/kg, K: 5 mg/kg, B1: 20 mg/kg, B2: 20 mg/kg, B6: 20 mg/kg, B12: 0.036 mg/kg, niacin: 100 mg/kg, folic acid: 6 mg/kg, biotin: 0.4 mg/kg, pantothenic acid: 60 mg/kg, and choline chloride: 1,500 mg/kg.

Reviewer #3: I am writing to submit my review report for the manuscript entitled “Live and heat-killed Leuconostoc mesenteroides counteract the gastrointestinal dysfunction in chronic kidney disease mice through intestinal environment modulation " for your consideration. Overall, I find the manuscript's findings intriguing and the information provided useful for researchers and academia. The article has the potential to make a significant contribution to the related discipline.

However, I have some concerns regarding the clarity, detail, and precision of different sections, which I outline below:

I recommend that the authors address these concerns and provide a revised version of the manuscript for further consideration

• Abstract-Keep a proper sequence in the abstract and end with conclusion. Focus on the gap you have covered in your study

• L-23 Abstarct- mention numerical values for better understanding>>CKD mice 23 treated with live and heat-killed Ln.m showed blood urea nitrogen and creatinine levels significantly 24 increased in the CKD compared to the control group, nevertheless, they were significantly reduced 25 in both live and heat-killed probiotic-treated groups. Kidney damage, Firmicutes/Bacteroidota 26 imbalance, increasing colonic uremic toxin, decreasing fecal short-chain fatty acids, and constipation 27 were observed in CKD.

• L-29 Abstract Reconsider and remove repetition - Taken together, Ln.m could be considered 29 a probiotic, and heat-killed Ln.m exhibits a similar effect to its live form in alleviating gastrointestinal 30 dysfunction and the progression of renal damage in CKD mic

We are grateful to the reviewer for his interest in our work and careful manuscript evaluation.

Following the reviewer’s recommendation, we have edited the abstract in the abstract part.

• L-59 Introduction – is it a reference statement???????????? As a result, there is 58 growing interest in strategies to manipulate and restore beneficial microbiota to improve health outcomes in the female 59 reproductive tra

We are not sure about the reviewer’ comment, because line 59 stated about constipation in CKD.

• L-47 Need little attention for better understanding - Notably, short-chain 47 fatty acids (SCFAs), such as acetic, propionic, and butyric acids, which are microbial metabolites, 48 decreased along with the decreasing diversity of gut microbiota in CKD mice (5). It was reported that 49 the alteration of gut microbiota and consequent reduction of SCFAs production played

Following the reviewer’s comment, we have rewritten the text in the manuscript. “Moreover, short-chain fatty acids (SCFAs), the key microbial metabolites such as acetic, propionic, and butyric acids were significantly reduced alongside the declining diversity of gut microbiota in CKD mice (Huang et al., 2021). Reduced SCFAs production has been identified as a critical factor contributing to intestinal dysmotility, including constipation, in CKD patients (Ikee et al., 2020). (see Page 3, line 49-53)

• L-58-check the cited reference- Prolonged intestinal transit has bidirectional effects with gut microbiota alteration that 59 result in increased gut-derived uremic toxins such as indole and p-Cresol which will be absorbed and 60 metabolized into indoxyl sulfate and p-Cresol sulfate, respectively. Indoxyl sulfate and p-Cresol 61 sulfate are normally excreted by the kidneys, but in CKD, the toxins accumulate in the circulation 62 and cause kidney fibrosis (6–8)

We have rechecked the references following the reviewer’s comment.

• L-92 You mean parabiotics???????????? ). Using inactivated probiotics could avoid the potential risks of live probiotics on 92 vulnerable or pediatric patients by translocating from the gastrointestinal tract to the blood of these 93 patients. In addition to the safety aspect, inactivated probiotics are easier to transport, store, and 94 standardize than live probiotics (‘

The inactivated probiotics are also known as parabiotics, paraprobiotics, or tyndallized probiotics.

• The introduction could be improved by providing more context and background from following latest references,

o doi: https://doi.org/10.1016/j.jep.2023.116503

o doi: https://doi.org/10.1016/j.ejphar.2024.176356

o https://doi.org/10.1016/j.lfs.2023.122380

o .doi: 10.3389/fnut.2024.1364841

Thank you for suggesting these potential references. We appreciate your recommendation and took the time to thoroughly review the article. However, we found that some references do not directly align with the scope or findings of our study. As such, we have decided to include some references for citation (Reference #42)

• L-105 Please elaborate the conditions for better understanding- To isolate the LAB from the gastrointestinal tract of Tiger prawn, the gastrointestinal tract 106 was collected under hygienic conditions. The sample was homogenized and incubated in the De Man, 107 Rogosa, and Sharpe (MRS) broth at 37℃ with shaking at 150 rpm for 24 h. It was plated on the MRS 108 agar and incubated at 37℃ for 24 h. in the

We elaborated the text in the manuscript for better understanding following the reviewer’s comment. “To isolate the LAB from different natural sources, fermented cabbage, fermented garlic, Sea bass's gastrointestinal content, and Tiger prawn's gastrointestinal content were collected under hygienic conditions. The samples were homogenized and incubated in the De Man, Rogosa, and Sharpe (MRS) broth at 37℃ with shaking at 150 revolutions per minute (RPM) for 24 h. Afterward, the serial dilution was performed before plating on the MRS agar and incubated at 37℃ for 24 h in the anaerobic condition. The colonies were selected from their different morphology”. (see Page 7, line 113-118)

• L-150 Check RPM- aerobic condition at 37ºC with shaking at 150 rpm for 24 h. The bacteria were harvested 143 by centrifugation at 8,000 rpm for 10 min and washed twice with 0. 85% NaCl. Bacterial cells were 144 resuspended in 0. 85% NaCl and then inactivated in the water bath at 100ºC for 10 min. The HK 145 bacteria were freeze- dried and kept at –80ºC until used. The freeze- dried bacteria powder was 146 suspended in sterile distilled water to be administer

We have checked the text from the reviewer’s comment and found that the RPM at 150 for shaking and at 8,000 for 10 min for harvesting the bacterial pellet was performed following the previous study (Somashekaraiah et al., 2019).

• L-159 need clarity ----After 28 days of induction, mice were administered treatments once a day for 28 days. 160 Freeze-dried live and HK probiotics were re-s

---

## [Editor Report · Decision Letter 1]

22 Jan 2025

Live and heat-killed Leuconostoc mesenteroides counteract the gastrointestinal dysfunction in chronic kidney disease mice through intestinal environment modulation

PONE-D-24-41563R1

Dear Dr. KHUITUAN,

We’re pleased to inform you that your manuscript has been judged scientifically suitable for publication and will be formally accepted for publication once it meets all outstanding technical requirements.

Kind regards,

Guadalupe Virginia Nevárez-Moorillón, Ph.D.

Academic Editor

PLOS ONE
---

## [Editor Report · Acceptance letter]

PONE-D-24-41563R1

PLOS ONE

Dear Dr. KHUITUAN,

I'm pleased to inform you that your manuscript has been deemed suitable for publication in PLOS ONE. Congratulations! Your manuscript is now being handed over to our production team.

Kind regards,

on behalf of

Dr. Guadalupe Virginia Nevárez-Moorillón

Academic Editor

PLOS ONE